# INSTANT: COMPRESSING GRADIENTS AND ACTIVATIONS FOR RESOURCE-EFFICIENT TRAINING

**Tuan-Kiet Doan,**\* **Trung-Hieu Tran,**\* **Enzo Tartaglione, Nikola Simidjievski , Van-Tam Nguyen**
LTCI, Télécom Paris, Institut Polytechnique de Paris, France
`tuan.doan@telecom-paris.fr, trung-hieu.tran@telecom-paris.fr,`
`enzo.tartaglione@telecom-paris.fr, nikola.simidjievski@telecom-paris.fr,`
`van-tam.nguyen@telecom-paris.fr`

## ABSTRACT

Deep learning has advanced at an unprecedented pace. This progress has led to a significant increase in its complexity. However, despite extensive research on accelerating inference, training deep models directly within a resource-constrained budget remains a considerable challenge due to its high computational and memory requirements. In this paper, we introduce INSTANT (compressIng gradieNtS and acTivAtions for resource-efficieNt Training), a method designed to address both the computational and the memory bottlenecks when training. INSTANT reduces resource demands during backpropagation by projecting gradients and activations into a low-rank subspace and performing computation within that compressed representation. Experimental results demonstrate that INSTANT achieves a $15\times$ reduction in computational cost and $32\times$ reduction in activation memory with negligible impact on model performance. The code is available at **INSTANT**.

## 1 INTRODUCTION

Deep learning has become the backbone of many practical applications in diverse fields such as computer vision (CV) (Dosovitskiy et al., 2020; Liu et al., 2021a), natural language processing (NLP) (Devlin et al., 2019; Liu et al., 2019; Radford et al., 2018), signal processing (Gong et al., 2021), and multimodal learning (Radford et al., 2021; Singh et al., 2022). Although these applications offer undeniable benefits, the large-scale design of deep learning models prevents their deployment on devices with limited resources. To address these shortcomings, research has mainly focused on two main directions. Many works focus on developing architecture modifications to adapt to hardware constraints (Sun et al., 2020; Li et al., 2022). Many others enhance quantization techniques to reduce memory footprint and improve inference speed for large models (Liu et al., 2021b), leveraging hardware that supports fast operations on low-bit datatypes. To summarize, the majority of the work focuses on model inference, while the **training part** is typically performed on an independent high-performance infrastructure.

Resource-efficient training faces two main challenges: high memory usage and computational cost. The memory overhead has been partially addressed in prior work (Nguyen et al., 2024; Yuan et al., 2023; Wang et al., 2025), typically employing singular value decomposition (SVD), to construct a low-rank space for activations and/or weights. However, these SVD-based methods incur substantial computational overhead, ultimately increasing training time (Nguyen et al., 2024). On the other hand, reducing computational cost during training remains an open area of research. (Sakr & Khailany, 2024) reduces the cost of tensor decomposition by constructing a periodically updated low-rank space for activation throughout the training process. (Yang et al., 2023b) leverages low-frequency characteristics of images to project tensors into predetermined low-rank spaces, which helps to save both activation memory and computational overhead for CV tasks. However, this method has a limited compression rate and is only effectively applicable to low-frequency data components such as images.

Inspired by tensor decomposition strategies, we propose INSTANT (compress**I**ng gradie**Nt**S and ac**T**iv**A**tions for resource-efficie**N**t **T**raining). INSTANT *reduces the resource demands of backpropagation, both in terms of computation and memory*, and applies to a wide range of data distributions.

---

\*Equal contribution.

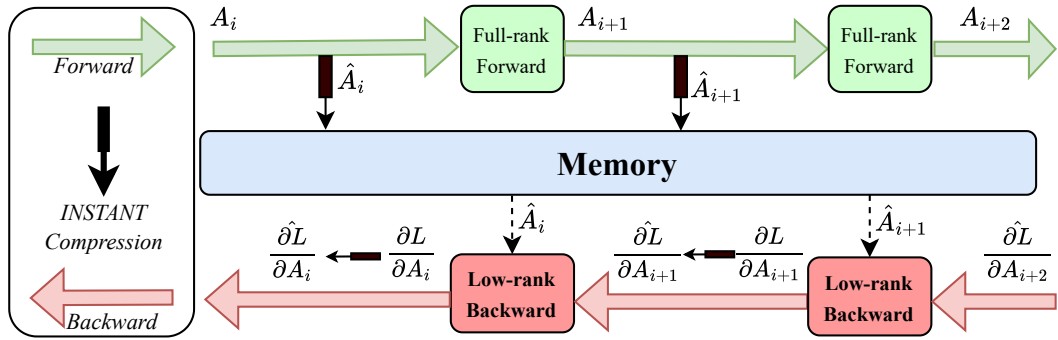

Figure 1: **INSTANT performs compression on activations and gradients**: In the forward pass (Green), activation $A_i$ normally propagates in full-rank space while its compact version $\hat{A}_i$ is saved for the backpropagation to reduce memory consumption. In the backward pass (Red), the gradient for the output $\frac{\partial L}{\partial A_{i+1}}$ is compressed to a low-rank version $\frac{\partial \hat{L}}{\partial A_{i+1}}$. Then, low-rank operations are implemented with compressed activation $\hat{A}_i$ and compressed gradient $\frac{\partial \hat{L}}{\partial A_{i+1}}$ to save backward computation.

Our method periodically identifies critical tensor features to generate dynamic low-rank projections, optimizing backpropagation efficiency. Our method is **orthogonal** to non-compressive acceleration techniques (Yu et al., 2022; Kwon et al., 2023), low-rank adaptation (Hu et al., 2022), low-rank parameter gradient (Zhao et al., 2024), and existing tensor compression techniques such as quantization (Xi et al., 2024). Our key contributions are as follows:

- We introduce a low-cost calibration technique to generate calibrated orthonormal bases for tensor projection, enabling significant reductions in memory and computations (Sec. 3.2).
- We project activation tensors and gradients onto these orthonormal bases. To our knowledge, this is the first work to exploit the low-rank structure of activation gradients for all types of data distribution. We provide an error analysis of our gradient compression, illustrating that a high compression ratio is achievable with limited performance degradation (Sec. 3.3).
- We evaluate INSTANT across multiple datasets and model architectures, consistently demonstrating good performance, achieving up to $32\times$ memory savings and $15\times$ computational cost reduction with only a $1\%$ trade-off in accuracy compared to vanilla fine-tuning (Sec. 4).

## 2 RELATED WORK

**Activation compression.** Activation compression is a recently emerging research direction that addresses the memory challenges during training. This approach offers several key advantages based on the following observations: (i) model weights remain uncompressed during training, thereby preserving their expressive capacity; (ii) activations are often large and exhibit significant redundancy, making them suitable for compression (Sakr & Khailany, 2024; Miles et al., 2024). (Nguyen et al., 2024) applies SVD to compress activations to reduce huge memory usage for activations. However, this approach raises substantial computational overhead due to the high cost of performing SVD in each training iteration. (Sakr & Khailany, 2024) (ESPACE) tackles SVD computational expense by using calibrated subspaces, which are periodically updated, to compress activations. They enable activation compression in the forward pass, reducing computational overhead in both the forward and backward phases. However, ESPACE is prone to error accumulation, as it relies on the universal fixed subspace across varying activations.

**Optimizer state compression.** Weight gradients are inherently low-rank (Yang et al., 2023a). Previous studies (Bernstein et al., 2018; Vogels et al., 2019) have leveraged this characteristic to address communication bottlenecks in distributed learning by reducing inter-device data transmission. GaLore (Zhao et al., 2024) and its variances (Muhamed et al., 2024; Shamshoum et al., 2025) leverage the low-rank property of weight gradients for compressing them to reduce memory usage in the optimizer state significantly. CompAct Shamshoum et al. (2025) further reduces the memory overhead

by compressing both optimizer state and activation memory. Nonetheless, in all aforementioned techniques, the activation gradient computations still rely entirely on high-cost backpropagation.

**Activation gradient compression.** Gradient filtering (Yang et al., 2023c) was proposed to pool both activation and gradient activation into a low-rank space for efficient computing. Since this technique helps increase efficiency, the performance drop is significantly large. (Yang et al., 2023b) proposes low-rank backpropagation via Walsh Hadamard transformation (LBP-WHT) to compress the gradient of output for multiplications in the low-rank space, which reduces the computational complexity. However, LBP-WHT is restricted to low-frequency inputs such as images, and achieves only modest compression ratios, resulting in limited gains in both memory efficiency and computational reduction. INSTANT, on the other hand, overcomes the low-frequency assumption via SVD, compressing the gradient into a smaller space compared to LBP-WHT, and as a result, can be applied to a variety of input types.

## 3  INSTANT

We start with discussing the computational and memory issues of vanilla backpropagation (Sec. 3.1). Next, we present our efficient construction of low-rank projectors for both activation and activation gradient in Sec. 3.2. Finally, we demonstrate our approach to perform low-rank multiplications in the backward by using these projectors, instead of full-rank computations (Sec. 3.3). Our objective here is to reduce both memory and computational consumption for the backpropagation.

### 3.1  PROBLEM STATEMENT

In the following, for simplicity, we focus on linear layers. Extension to convolutional layers is presented in Appendix H. Within each layer, the batch dimension and bias are omitted without loss of generality. Let $\mathbf{x} \in \mathbb{R}^{L \times C_x}$ denote the input activation, $\mathbf{y} \in \mathbb{R}^{L \times C_y}$ denote the output activation, and $\mathbf{w} \in \mathbb{R}^{C_y \times C_x}$ denote the weight matrix, where $L$ denote the sequence length and $C_x$ and $C_y$ are layer channel dimensions, which are predefined by model architecture.

In the forward phase, the input $\mathbf{x}$ is propagated through the layer to compute the output: $\mathbf{y} = \mathbf{x} \cdot \mathbf{w}^\top$. In supervised learning, the output of the final layer is compared against the ground truth label to calculate a loss value $\mathcal{L}$. The backward pass, also known as backpropagation, computes the gradients of this loss to input $\mathbf{x}$ and weight $\mathbf{w}$. We denote $\mathbf{g_y} = \frac{\partial \mathcal{L}}{\partial \mathbf{y}} \in \mathbb{R}^{L \times C_y}$ as gradient of the loss with respect to the output $\mathbf{y}$. Gradient of the loss with respect to the input $\mathbf{x}$, and the weight $\mathbf{w}$ are denoted as $\mathbf{g_x} = \frac{\partial \mathcal{L}}{\partial \mathbf{x}}$ and $\mathbf{g_w} = \frac{\partial \mathcal{L}}{\partial \mathbf{w}}$, respectively, calculated as:

$$\mathbf{g_x} = \mathbf{g_y} \cdot \mathbf{w}, \quad \mathbf{g_w} = \mathbf{g_y}^\top \cdot \mathbf{x}. \tag{1}$$

Here, $\mathbf{g_w} \in \mathbb{R}^{C_y \times C_x}$ is used to update the weight $\mathbf{w}$, and $\mathbf{g_x} \in \mathbb{R}^{L \times C_x}$ is propagated to the preceding layer.

**Training overhead.** The backward pass ($equation\ 1$) includes two matrix multiplications, each with the same cost, resulting in a total of $4 \cdot L \cdot C_x \cdot C_y$ floating point operations (FLOPs). In Transformer-based models, the values of $L$, $C_x$, and $C_y$ are typically large, leading to a computational burden, especially during the backward stage. Moreover, since the backward pass requires storing the input activation $\mathbf{x}$ for computing $\mathbf{g_w}$, memory consumption can significantly increase.

To address the dual problems of high memory and computation, we propose a tensor decomposition strategy for both input activation $\mathbf{x}$ and the output gradient $\mathbf{g_y}$. As shown in Fig. 1, we project the activation $\mathbf{x}$ and gradient $\mathbf{g_y}$ into a lower-dimensional space for storage and computation. Unlike previous works (Yang et al., 2023b) that use universal projections for both of these tensors, we adopt adaptive projections that better capture the crucial information of each tensor. Next, we discuss our construction scheme of these low-rank projections for each layer.

### 3.2  EFFICIENT CONSTRUCTION FOR TENSOR PROJECTION

Low-rank and low-cost projections are required to effectively compress the input activation $\mathbf{x}$ and the output gradient $\mathbf{g_y}$. Previous approaches (Nguyen et al., 2024) use SVD with an operational cost of $\mathcal{O}(n^3)$ to compute these subspaces at every training step, making them impractical for large-scale training.

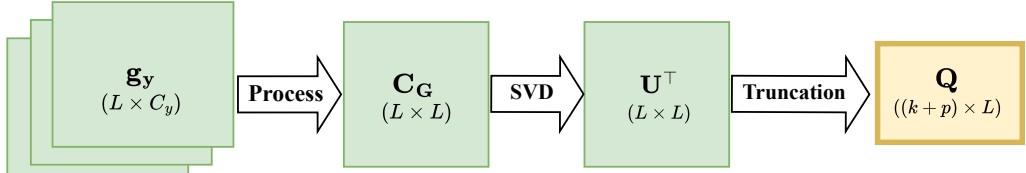

(a) Constructing compression tensors $\mathbf{Q}$ for output gradient $\mathbf{g_y}$. Similarly, $\mathbf{P}$ is constructed for activation $\mathbf{x}$.

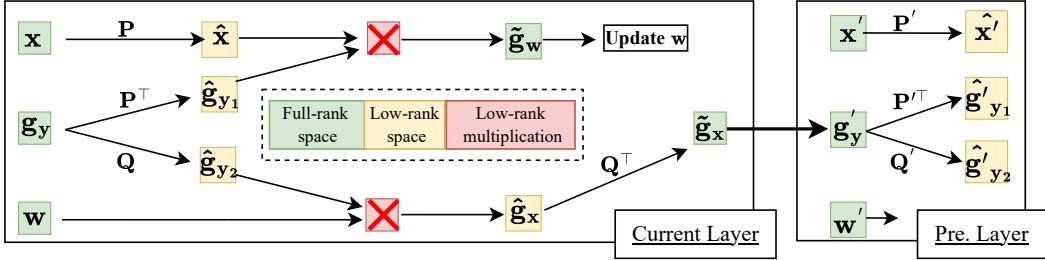

(b) Low-rank Backpropagation algorithm with INSTANT. Instead of **full-rank multiplication** as Vanilla, INSTANT decomposes $\mathbf{x}, \mathbf{g_y}$ to approximate $\mathbf{g_w}$ and $\mathbf{g_x}$ by **low-rank multiplications**.

Figure 2: INSTANT backpropagation involves calibrated low-rank tensors $\mathbf{P}, \mathbf{Q}$, which are updated in calibration process. These low-rank projections reduce memory consumption of saving activation $\mathbf{x}$ and reduce computations thanks to low-rank multiplications.

To alleviate this overhead, we adopt a calibrating strategy in which we create periodically updated compression subspaces for compressing activation $\mathbf{x}$ and output gradient $\mathbf{g_y}$ in substantial $N_t$ training steps. For good approximations, each tensor demands its specific compression (Appendix. B.1). Our approach is inspired by ESPACE's theoretical foundation (Sakr & Khailany, 2024) for a fixed subspace to compress the activation. Given activation tensor $\mathbf{x} \in \mathbf{X}$, by performing SVD on activation auto-correlation: $\mathbf{C_X} = \mathbb{E}[\mathbf{x} \cdot \mathbf{x}^\top] = \mathbf{U\Sigma U}^\top$, where $\mathbf{U}$ are singular vectors and $\mathbf{\Sigma}$ are their associated singular values, the projection matrix built from $\mathbf{U}$ minimizes mean square error (MSE) of decomposing activation $\mathbf{x}$ (Appendix. B.2). Different from ESPACE, we decompose both activation $\mathbf{x}$ and output gradient $\mathbf{g_y}$, and do not stretch their batch dimension for these decompositions.

**Data preprocessing.** To approximate auto-correlation tensors, many data batches are required to get sufficient statistics of activation $\mathbf{x}$ and output gradient $\mathbf{g_y}$. In this calibration stage, if we naively save all of this data for post-processing, it will lead to memory accumulation, which is at odds with the goal of reducing memory. Therefore, we implement low-cost data preprocessing to limit its storage demand below that of the training phase, ensuring *low-computational calibration and peak memory usage does not increase* (Appendix. B.3). For simplicity, in this section, we present the approach to find the compression subspace $\mathbf{Q}$ for gradient $\mathbf{g_y}$ (Fig. 2a). A similar strategy is applied for finding a compression subspace $\mathbf{P}$ of activation $\mathbf{x}$.

**Singular value decomposition.** After getting output gradient auto-correlation $\mathbf{C_G} = \mathbb{E}(\mathbf{g_y} \cdot \mathbf{g_y}^\top)$ via data preprocessing, we decompose this to get the left singular vectors $\mathbf{U}$ and their associated singular values $\mathbf{\Sigma}$ in decreasing order. Projection matrix built from $\mathbf{U}$ is proved to minimize MSE of decomposing $\mathbf{g_y}$.

**Truncation with energy threshold.** To reduce dimensionality while preserving information, we define energy threshold $\epsilon$ ($\epsilon \leq 1$) as the portion of tensor energy remaining after decomposition. We define $\mathcal{E}$ as the sum of squares of all eigenvalues $\sigma_i$ in $\mathbf{\Sigma}$:

$$\mathcal{E} = \sum \sigma_i^2 = \|\mathbf{C_G}\|_\mathbf{F}^\mathbf{2}, \tag{2}$$

where $\|\mathbf{C_G}\|_\mathbf{F}^\mathbf{2}$ is the squared Frobenius norm of $\mathbf{C_G}$. Following that, we truncate $k$ vectors of $\mathbf{U}$ to form $\mathbf{U_k}$, where truncation index $k$ is the minimal integer satisfying: $\sum_{i=1}^{k} \sigma_i^2 \geq \epsilon \cdot \mathcal{E}$. When $\epsilon \to 1$, this truncation strategy preserves most of tensor energy, hence assigning $\mathbf{Q} = \mathbf{U_k}^\top$ reduces decomposition MSE between $\mathbf{g_y}$ and its reconstruction $\tilde{\mathbf{g}}_\mathbf{y} = \mathbf{Q}^\top \cdot \mathbf{Q} \cdot \mathbf{g_y}$. Compression tensor $\mathbf{Q}$ is extremely

smaller than $\mathbf{g_y}$ due to its low-rank characteristic (as shown in Sec. 4.2), therefore, $\mathbf{Q}$ can reduce memory and computational expense when joining backward operations (Fig. 2b).

**Truncation with Oversampling.** The energy threshold ensures that a certain amount of information is preserved when the calibration happens. However, as training progresses, this projection may no longer suffice to maintain that amount of information, since the core bases on which to project can vary. To address this, we propose oversampling, which increases the number of base projections by a fixed amount, thus reducing information loss when the core bases change (Fig. 2a). This truncation technique is proven to be effective in Sec. 4.4 and further investigated in Appendix. F.

Dropping lower singular values leads to an accumulation of energy loss when backpropagating reconstructed tensors. We, therefore, propose an energy offset mechanism to compensate for this loss at each truncation. Finally, $\mathbf{Q}$ is given as:

$$\mathbf{Q} = \mathbf{U_{k+p}^\top} \cdot \left( \sum_{i=1}^{k+p} \sigma_i^2 \right)^{-\frac{1}{2}}, \qquad (3)$$

where $\mathbf{Q} \in \mathbb{R}^{R_y \times L}$, with $R_y = k + p$. Applying the similar strategy, we find compression subspace $\mathbf{P} \in \mathbb{R}^{R_x \times L}$ for low-rank activation $\mathbf{x}$.

### 3.3 LOW-RANK BACKPROPAGATION WITH INSTANT

The activation and gradient can be mapped onto low-rank spaces with two projection tensors constructed in Sec. 3.2. This part will demonstrate the use of these low-rank matrices $\mathbf{P}$ and $\mathbf{Q}$ in our training process. Our overall training pipeline is depicted in Fig. 1, while the detailed technique for handling the low-rank backward is illustrated in Fig. 2b.

Since changing the forward pass of the network can cause a significant performance drop, we keep the forward pass $\mathbf{y} = \mathbf{x} \cdot \mathbf{w}^\top$ unchanged. Meanwhile, the activation map $\mathbf{x}$ is projected into a smaller space: $\hat{\mathbf{x}} = \mathbf{P} \cdot \mathbf{x}$ where $\mathbf{P} \in \mathbb{R}^{R_x \times L}$, $\hat{\mathbf{x}} \in \mathbb{R}^{R_x \times C_x}$ with $R_x \ll L$. This compressed activation $\hat{\mathbf{x}}$ is retained in memory in place of $\mathbf{x}$ for the backward pass, which saves a large amount of memory.

The backpropagation process of INSTANT is described in Fig. 2b. By the property of the low-rank projection tensor $\mathbf{P}$ built in Sec. 3.2: $\mathbf{x} \approx \mathbf{P}^\top \cdot \mathbf{P} \cdot \mathbf{x}$, weight gradient is approximated:

$$\mathbf{g_w} = \mathbf{g_y^\top} \cdot \mathbf{x} \approx \mathbf{g_y^\top} \cdot (\mathbf{P}^\top \cdot \mathbf{P} \cdot \mathbf{x}) = (\mathbf{g_y^\top} \cdot \mathbf{P}^\top) \cdot (\mathbf{P} \cdot \mathbf{x}) \qquad (4)$$

This reordering results $\mathbf{g_w} \approx \tilde{\mathbf{g}}_\mathbf{w} = \hat{\mathbf{g}}_{\mathbf{y_1}} \cdot \hat{\mathbf{x}}$ with 2 low-dimensional spaces $\hat{\mathbf{g}}_{\mathbf{y_1}} \in \mathbb{R}^{R_x \times C_y}$ and $\hat{\mathbf{x}} \in \mathbb{R}^{R_x \times C_x}$ with $R_x \ll \min(L, C_x, C_y)$, as illustrated in Fig. 2b.

Similarly, by the property of low-rank projection tensor $\mathbf{Q}$ built in Sec. 3.2: $\mathbf{g_y} \approx \mathbf{Q}^\top \cdot \mathbf{Q} \cdot \mathbf{g_y}$, input gradient is approximated with 3 low-rank multiplications:

$$\hat{\mathbf{g}}_{\mathbf{y_2}} = \mathbf{Q} \cdot \mathbf{g_y}, \qquad \hat{\mathbf{g}}_\mathbf{x} = \hat{\mathbf{g}}_{\mathbf{y_2}} \cdot \mathbf{w}, \qquad \tilde{\mathbf{g}}_\mathbf{x} = \mathbf{Q}^\top \cdot \hat{\mathbf{g}}_\mathbf{x}. \qquad (5)$$

where $\hat{\mathbf{g}}_{\mathbf{y_2}} \in \mathbb{R}^{R_y \times C_y}$, $\hat{\mathbf{g}}_\mathbf{x} \in \mathbb{R}^{R_y \times C_x}$ (2 low-rank spaces). Finally, the approximated input gradient $\tilde{\mathbf{g}}_\mathbf{x} \in \mathbb{R}^{L \times C_x}$, is propagated to the preceding layer.

Equation 4 and equation 5 will cause error compared to traditional backpropagation. **This error is mathematically proved to be negligible with our low-rank projection scheme** (Appendix. C).

**Training overhead.** In total, the computational cost of INSTANT for backpropagating one layer, which includes low-rank compression, low-rank computation, and reverse projection, is $2 \cdot (R_x + R_y) \cdot (C_x \cdot C_y + L \cdot C_x + L \cdot C_y)$ FLOPs as shown in the Appendix. D.1. Given that $R_x + R_y \ll \min(L, C_x, C_y)$, this number is much smaller than $4 \cdot L \cdot C_x \cdot C_y$ of vanilla training. For example, one BERT block has $L = 512, C_x = C_y = 768$, choosing $R_x = R_y = 8$ can save about $27\times$ FLOPs. Remarkably, $\mathbf{P}$, $\mathbf{Q}$, and $\hat{\mathbf{x}} = \mathbf{P} \cdot \mathbf{x}$ are only used during training, meaning that at inference time, there is no trade-off in memory or computation compared to traditional inference. Moreover, INSTANT focuses solely on reducing computational and memory costs during backpropagation, without modifying the optimizer state. INSTANT is orthogonal and can be combined with any technique that compresses the optimizer state.

# 4 RESULTS

## 4.1 EXPERIMENTAL SETUP

**Computer vision tasks.** We conduct experiments for image classification with a similar setup to LBP-WHT (Yang et al., 2023b). We assess our method on ImageNet (Russakovsky et al., 2015)-pretrained Vision Transformer models (EfficientFormer-L1 (Li et al., 2022), EfficientFormerV2-S0 (Li et al., 2023), and SwinV2-Small (Liu et al., 2022)) by fine-tuning them on five different datasets for 50 epochs. Details of the datasets are discussed in the Appendix. We use a batch size of 64 and use AdamW optimizer with the same learning rate schedules as in (Yang et al., 2023b). To balance training efficiency with performance, the calibration process is carried out every $N_t = 200$ iterations, and an energy threshold $\epsilon = 95\%$ is kept constant throughout all experiments, with only the oversampling hyperparameter $p$ being varied. In Tab. 1 and Tab. 2, we denote INSTANT-$p$ as our method with energy threshold of $\epsilon = 95\%$ and oversampling $p$ vectors. For fine-tuning the last layer, $p$ is tested with values of $0, 5$, and $7$, while for fine-tuning the entire model, $p$ is tested with values of $5, 10$, and $15$.

**Natural language processing tasks.** We employ two Transformer-based models (BERT (Devlin et al., 2019) and DistilBERT (Sanh et al., 2019)), which are pretrained on Wikipedia and BookCorpus. We utilise the GLUE benchmark (Wang et al., 2018) to evaluate model performance, including 6 datasets: CoLA, QNLI, MRPC, RTE, SST-2, and MNLI. For each dataset, each model is fine-tuned for 10 epochs with the AdamW optimizer and a batch size of 32. To balance between training efficiency and performance, the calibration process is carried out after each $N_t = 50$ iterations, and we maintain an energy threshold $\epsilon = 95\%$ across all experiments, adjusting only the oversampling hyperparameter $p$. For fine-tuning the last layer, we test $p$ with values of $0, 7$, and $15$, while for fine-tuning the entire model, $p$ is tested with values of $7, 15$, and $25$.

**Baseline.** Besides vanilla fine-tuning (Vanilla in Tab. 1, 2), we also reproduce and evaluate 2 other methods: Gradient Filter (Yang et al., 2023c) and LBP-WHT (Yang et al., 2023b). For image classification tasks, we strictly follow the authors' setup and configuration, whereas for language tasks, we reshape the sequence length of each input into an appropriate two-dimensional tensor to apply their method. We use the notation LBP-WHT-$o$ like the authors, with $o$ denoting the order that determines the rank of the compressing subspace. We compare each method based on average accuracy (mAcc), average Mega FLOPs (MFLOPs), and average activation memory consumption (Mem). Notably, FLOPs and memory are only reported for the Linear layers, which are the heaviest computational components in these architectures. We measure computational cost in FLOPs rather than time, as it's unaffected by implementation details. Therefore, this metric allows us to evaluate the efficiency gains from a better algorithm rather than implementation aspects. All experiments are performed on a NVIDIA TESLA V100, and the source code uses PyTorch 1.13.1. We use the MMCV library (Contributors, 2018) for CV tasks and use the Hugging Face library (Wolf et al., 2020) for NLP tasks.

## 4.2 THE ACTIVATION GRADIENT IS LOW-RANK

We conduct experiments with BERT on the MRPC dataset. In the fine-tuning process, we randomly select samples and track their gradients corresponding to the output at each layer. SVD is then applied to each gradient to extract its eigenvalues. As shown in Fig. 3, the number of ranks required to keep $\epsilon = 95\%$ of energy is only 6. Since most of the energy is concentrated in a few top eigenvalues, it suggests that a small number of ranks can retain a significant portion of the gradient's information. We have validated this phenomenon in a wider range of samples and layers. Further details for other layers in other architectures are provided in the Appendix K.5. This observation supports our idea of projecting the activation gradient into a smaller subspace, where we perform computations to reduce computational cost while preserving a large amount of the gradient's information.

## 4.3 MAIN RESULTS

**Computer vision tasks.** Tab. 1 presents the results of INSTANT in comparison with vanilla fine-tuning and other gradient and activation compression methods on Vision Transformer models. It is noticeable that INSTANT achieves a significant reduction in both MFLOPs and memory usage compared to Vanilla. In EfficientFormer, INSTANT-0 witnesses drops from $2\%$ to $5\%$ compared to

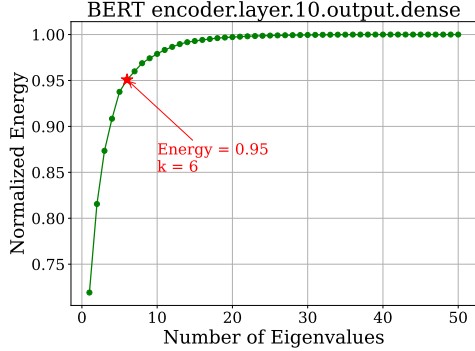

Figure 3: The percentage of energy retained depends on the number of eigenvalues in one layer in BERT.

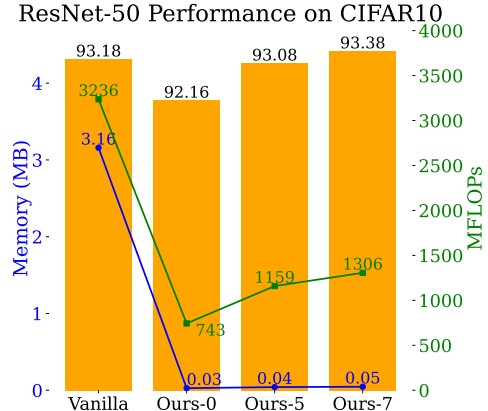

Figure 4: Fine-tuning ResNet-50 on CIFAR10

Table 1: Experimental results across 5 different CV datasets, presented for both fine-tuning the last block and the entire model. We report the MFLOPs and memory (Mem) required for training a single sample. Detailed measurement methodologies can be found in the Appendix. D.1, D.2, D.3

| Model | Method | MFLOPs ↓ | Mem (MB) ↓ | CF100 ↑ | CF10 ↑ | Flowers ↑ | Food ↑ | Pets ↑ | mAcc ↑ |
|---|---|---|---|---|---|---|---|---|---|
| **Fine-tuning the Last Block** | | | | | | | | | |
| Efficient Former-L1 | Vanilla | 1484 | 1.95 | 79.28 | 95.23 | 95.50 | 84.04 | 93.13 | 89.44 |
| | Gradient Filtering | 24 | 0.04 | 68.29 | 90.72 | 90.01 | 74.61 | 88.93 | 82.51 |
| | LBP-WHT-2 | 95 | 0.12 | 75.61 | 93.35 | 95.07 | 79.65 | 92.34 | 87.20 |
| | LBP-WHT-4 | 335 | 0.40 | 78.27 | 94.6 | 95.53 | 82.37 | 93.16 | 88.79 |
| | LBP-WHT-8 | 1227 | 1.43 | 79.34 | 95.31 | 95.58 | 83.98 | 92.94 | **89.43** |
| | INSTANT-0 | 270 | 0.16 | 77.64 | 94.66 | 92.23 | 81.97 | 92.64 | 87.83 |
| | INSTANT-5 | 475 | 0.38 | 78.65 | 95.07 | 95.93 | 82.84 | 93.21 | 89.14 |
| | INSTANT-7 | 544 | 0.45 | 79.01 | 95.23 | 95.92 | 83.05 | 93.02 | 89.25 |
| Efficient FormerV2-S0 | Vanilla | 349 | 1.47 | 72.37 | 92.63 | 92.73 | 81.44 | 90.52 | 85.94 |
| | Gradient Filtering | 7 | 0.03 | 64.17 | 87.03 | 87.9 | 73.67 | 85.55 | 79.66 |
| | LBP-WHT-2 | 28 | 0.09 | 65.75 | 88.68 | 89.51 | 74.72 | 87.49 | 81.23 |
| | LBP-WHT-4 | 99 | 0.30 | 69.03 | 90.88 | 90.73 | 79.45 | 89.29 | 83.88 |
| | LBP-WHT-8 | 363 | 1.08 | 71.90 | 92.29 | 92.60 | 81.07 | 90.52 | **85.68** |
| | INSTANT-0 | 119 | 0.03 | 64.51 | 89.62 | 87.94 | 76.77 | 88.12 | 81.39 |
| | INSTANT-5 | 161 | 0.15 | 68.61 | 90.85 | 90.86 | 79.07 | 88.83 | 83.64 |
| | INSTANT-7 | 181 | 0.20 | 69.42 | 91.05 | 90.52 | 79.52 | 88.88 | 83.88 |
| SwinV2 Small | Vanilla | 2718 | 2.25 | 80.84 | 96.07 | 97.61 | 88.31 | 95.53 | 91.67 |
| | Gradient Filtering | 37 | 0.04 | 80.19 | 95.68 | 97.4 | 87.31 | 94.9 | 91.10 |
| | LBP-WHT-2 | 141 | 0.12 | 80.23 | 95.65 | 97.50 | 88.06 | 94.47 | 91.18 |
| | LBP-WHT-4 | 499 | 0.41 | 80.39 | 95.71 | 97.54 | 88.32 | 94.66 | 91.32 |
| | LBP-WHT-8 | 1830 | 1.48 | 80.80 | 95.80 | 97.56 | 88.19 | 95.07 | 91.48 |
| | INSTANT-0 | 181 | 0.05 | 80.55 | 95.69 | 97.43 | 88.30 | 95.12 | 91.42 |
| | INSTANT-5 | 435 | 0.26 | 80.85 | 95.91 | 97.38 | 88.42 | 95.20 | 91.55 |
| | INSTANT-7 | 530 | 0.33 | 80.98 | 95.96 | 97.45 | 88.29 | 95.23 | **91.58** |
| **Full fine-tuning** | | | | | | | | | |
| Efficient Former-L1 | Vanilla | 4528 | 18.46 | 84.84 | 96.99 | 94.84 | 85.64 | 93.16 | 91.09 |
| | Gradient Filtering | 90 | 0.34 | 41.05 | 75.22 | 69.41 | 41.49 | 61.00 | 57.63 |
| | LBP-WHT-4 | 1211 | 3.36 | 77.97 | 94.13 | 93.38 | 41.88 | 92.23 | 79.92 |
| | LBP-WHT-6 | 2560 | 7.05 | 83.00 | 96.45 | 94.58 | 83.77 | 92.86 | 90.13 |
| | LBP-WHT-8 | 4400 | 12.08 | 83.88 | 96.78 | 94.70 | 85.08 | 93.32 | **90.75** |
| | INSTANT-5 | 2107 | 1.98 | 82.41 | 96.29 | 94.50 | 83.87 | 92.78 | 89.97 |
| | INSTANT-10 | 2491 | 2.73 | 83.05 | 96.48 | 94.73 | 84.66 | 93.13 | 90.41 |
| | INSTANT-15 | 2884 | 3.45 | 83.56 | 96.85 | 94.47 | 84.85 | 93.1 | 90.57 |

Vanilla, especially in high variance datasets such as CF100, Flowers, and Foods. This is possibly because the strict energy threshold approach in INSTANT-0 is not efficient with varying distributions of these datasets. INSTANT-5 and INSTANT -7 successfully address this issue, supporting our hypothesis that considering only tensor energy is insufficient. Compared to LBP-WHT, in EfficientFormer-L1 and SwinV2, INSTANT-5 gains comparable performance to LBP-WHT-8 with only $25\%$ computation and $18\%$ memory consumption. In the full-finetuning, INSTANT-10 outperforms LBP-WHT-6, and

Table 2: Experimental results on GLUE benchmark, presented for both fine-tuning the last layer and the entire model. We report the MFLOPs and memory (Mem) required for training a single sample. Detailed measurement methodologies can be found in the Appendices D.1, D.2, D.3

| Model | Method | MFLOPs ↓ | Mem (MB) ↓ | Datasets | | | | | | mAcc ↑ |
|---|---|---|---|---|---|---|---|---|---|---|
| | | | | MRPC ↑ | CoLA ↑ | QNLI ↑ | RTE ↑ | SST-2 ↑ | MNLI ↑ | |
| **Fine-tuning the Last Block** | | | | | | | | | | |
| BERT | Vanilla | 14495 | 13.50 | 83.92 | 43.55 | 86.12 | 59.21 | 91.63 | 78.31 | 73.79 |
| | Gradient Filter | 226 | 0.21 | 82.16 | 40.97 | 75.87 | 58.48 | 88.30 | 63.12 | 68.15 |
| | LBP-WHT-2 | 732 | 0.63 | 82.21 | 41.09 | 79.32 | 59.21 | 89.11 | 68.17 | 69.85 |
| | LBP-WHT-4 | 2464 | 2.11 | 82.33 | 42.46 | 83.82 | 58.84 | 90.71 | 73.62 | 71.96 |
| | INSTANT-0 | 175 | 0.03 | 82.21 | 41.35 | 79.33 | 61.73 | 90.71 | 64.39 | 69.95 |
| | INSTANT-7 | 565 | 0.21 | 82.33 | 41.58 | 84.13 | 62.09 | 90.94 | 72.92 | 72.33 |
| | INSTANT-15 | 1018 | 0.42 | 83.31 | 43.02 | 84.68 | 61.01 | 91.63 | 74.68 | **73.06** |
| Distil-BERT | Vanilla | 14495 | 13.50 | 82.55 | 35.52 | 82.85 | 59.21 | 88.99 | 73.60 | 70.45 |
| | Gradient Filter | 226 | 0.21 | 82.46 | 32.80 | 73.99 | 58.48 | 85.89 | 54.31 | 64.66 |
| | LBP-WHT-2 | 732 | 0.63 | 82.58 | 33.54 | 76.86 | 58.12 | 87.61 | 60.43 | 66.52 |
| | LBP-WHT-4 | 2464 | 2.11 | 82.46 | 31.73 | 80.62 | 56.68 | 89.11 | 67.56 | 68.03 |
| | INSTANT-0 | 160 | 0.03 | 82.14 | 29.89 | 74.81 | 58.84 | 88.30 | 62.42 | 66.07 |
| | INSTANT-7 | 546 | 0.21 | 82.41 | 33.50 | 80.49 | 59.57 | 89.56 | 68.52 | **69.01** |
| | INSTANT-15 | 999 | 0.42 | 82.41 | 32.96 | 80.93 | 58.12 | 88.99 | 69.68 | 68.85 |
| **Full fine-tuning** | | | | | | | | | | |
| BERT | Vanilla | 173946 | 162 | 90.23 | 58.69 | 91.43 | 67.51 | 93.23 | 84.46 | 80.76 |
| | Gradient Filter | 2716 | 2.53 | 84.49 | 43.02 | 81.62 | 64.98 | 87.96 | 68.60 | 71.78 |
| | LBP-WHT-2 | 8784 | 7.59 | 86.64 | 48.60 | 84.14 | 63.90 | 89.11 | 70.91 | 73.88 |
| | LBP-WHT-4 | 29579 | 25.31 | 87.80 | 46.12 | 86.16 | 65.70 | 90.94 | 77.32 | 75.67 |
| | INSTANT-7 | 9143 | 2.83 | 87.93 | 57.71 | 89.66 | 64.62 | 92.22 | 81.36 | 78.92 |
| | INSTANT-15 | 15353 | 5.43 | 87.97 | 58.08 | 90.63 | 62.82 | 92.43 | 83.06 | 79.17 |
| | INSTANT-25 | 20753 | 8.52 | 89.47 | 57.29 | 90.79 | 63.54 | 93.35 | 83.45 | **79.65** |

compared to Vanilla, it achieves $1.8\times$ computational reduction and $6.3\times$ memory reduction on average, at the expense of $0.8\%$ accuracy drop. It is noticeable that LBP-WHT-8 gains good performance at the expense of a small compression rate, especially in computation. Meanwhile, although the Gradient Filter can save a large amount of computational resources, the performance drops remarkably. This shows that this method cannot keep enough information for low-rank projection.

**Natural language processing tasks.** Tab. 2 illustrates INSTANT's effectiveness on 6 datasets of the GLUE benchmark. In fine-tuning the last block, compared to Vanilla, INSTANT-15 can save $14.37\times$ FLOPs and $32.14\times$ in memory, with a small reduction in accuracy of $1.17\%$. Notably, the difference in mAcc mainly comes from the MNLI dataset. This may be due to the large distribution of the dataset, which causes the method to struggle in finding an appropriate low-rank space. We believe that performance on this dataset can be further improved by increasing the oversampling amount. Compared to LBP-WHT, in BERT, INSTANT-7 achieves a $2.48\%$ accuracy boost over LBP-WHT-2, while it requires only about $75\%$ of the FLOPs and $33\%$ of the memory. In the full fine-tuning, INSTANT shows a slight performance decline relative to the Vanilla baseline. In contrast, LBP-WHT shows larger performance degradation across various datasets. This indicates that the low-frequency transformation assumption may not hold well for language tasks. INSTANT, with its SVD-based projection, can mitigate this issue to some extent, resulting in better performance compared to LBP-WHT. Note that calibration computational overhead is excluded from Tab.1, Tab.2. This overhead is proven to be much smaller than training overhead in Appendix B.3.

**Extension to convolution.** INSTANT is applicable to convolutional layers (as shown in Appendix. H). We conduct experiments with MobileNetV2 and ResNet-50 architectures on CIFAR10 and CIFAR100. As indicated in Fig.4, INSTANT can achieve a small accuracy boost with $3\times$ computation savings and $79\times$ memory savings, compared to Vanilla. Extra results are indicated in Fig.9. These results demonstrate the efficiency of INSTANT across all architectures, including Transformer- and Convolutional-based models.

**Edge device latency.** We conduct experiments on Raspberry Pi 5 (CPU ARM Cortex-A76). We report the average training time over 1 epoch on CIFAR10. The experimental setup is the same as the one indicated as Sec 4.1. By saving computations, INSTANT reduces $2\times$ of backward time, which decreases the average total training time, compared to Vanilla, as shown in Fig. 5. The time reduction is up to $12\times$ in another architecture, as indicated in the Appendix. I.1. However, INSTANT may increase GPU latency, as shown in the Appendix. I.2.

Table 3: Comparison of INSTANT with GaLore and CompAct, presented for both fine-tuning the last layer and the entire model. We report the MFLOPs and memory (Mem) required for training a single sample on QNLI and SST2.

| Model | Method | MFLOPs ↓ | Mem (MB) ↓ | QNLI ↑ | SST-2 ↑ |
|-------|--------|----------|------------|--------|---------|
| **Fine-tuning the Last Block** | | | | **Datasets** | |
| BERT | Vanilla | 14495 | 13.50 | 86.12 | 91.63 |
| | GaLore-8 | 14495 | 13.50 | 82.45 | 90.47 |
| | GaLore-32 | 14495 | 13.50 | 84.37 | 91.28 |
| | CompAct-8 | 7978 | 0.11 | 80.65 | 88.30 |
| | CompAct-32 | 8355 | 0.44 | 84.37 | 91.28 |
| | INSTANT-0 | 175 | 0.03 | 79.33 | 90.71 |
| | INSTANT-7 | 565 | 0.21 | 84.13 | 90.94 |
| | INSTANT-15 | 1018 | 0.42 | **84.68** | **91.63** |
| **Full fine-tuning** | | | | | |
| BERT | Vanilla | 173946 | 162 | 91.43 | 93.23 |
| | GaLore-8 | 173946 | 162 | 90.44 | 91.74 |
| | GaLore-32 | 173946 | 162 | **91.31** | 92.09 |
| | CompAct-8 | 95736 | 1.32 | 89.09 | 91.28 |
| | CompAct-32 | 100260 | 5.28 | 90.30 | 92.09 |
| | INSTANT-0 | 9143 | 2.83 | 89.66 | 92.22 |
| | INSTANT-15 | 15353 | 5.43 | 90.63 | 92.43 |
| | INSTANT-25 | 20753 | 8.52 | 90.79 | **93.35** |

**Comparison of INSTANT with optimizer state compression techniques.** We provide additional results to compare INSTANT with two optimizer compression techniques, which are GaLore (Zhao et al., 2024) and CompAct (Shamshoum et al., 2025), as presented in Tab. 3. GaLore consistently achieves better performance than CompAct under the same low-rank constraint; however, its activation memory consumption and computational cost remain similar to Vanilla training. In contrast, CompAct substantially reduces activation memory by compressing activations during the forward pass. Compared to CompAct, under the same activation memory budget, INSTANT can save a large portion of the backward computational cost while achieving better performance.

## 4.4 Ablation study

**Compressing both components has practical benefits.** We conduct ablation studies of *compressing only* $\mathbf{g_y}$ (INSTANT (compress $\mathbf{g_y}$)) and *compressing only* $\mathbf{x}$ (INSTANT (compress $\mathbf{x}$)); presented in Tab. 4). *Compressing only* $\mathbf{g_y}$) achieves good performance, even surpassing Vanilla training. However, due to saving full-rank activation, INSTANT (compress $\mathbf{g_y}$) requires the same memory as Vanilla. On the other hand, *compressing only* $\mathbf{x}$ achieves good performance with a high compression rate, which saves a large amount of memory. However, this method incurs a low compression rate in computation, because it involves one full-rank multiplication in the backward pass. INSTANT (compressing both $\mathbf{x}$, $\mathbf{g_y}$) achieves the highest computation compression rate and high memory compression rate with a negligible performance drop compared to Vanilla. In short, INSTANT (only compress $\mathbf{x}$) is a good choice if tasks only require a small memory budget. INSTANT (compressing both $\mathbf{x}$, $\mathbf{g_y}$) is good at tasks requiring a small memory budget and small training time, making it efficient for resource-efficient training.

**Energy is not enough.** The energy threshold $\epsilon$ is a local measure for truncating the compression tensor, which is based completely on the calibration data. This hyperparameter does not account for changes in the low-rank space during training, potentially leading to a performance drop. As shown in Fig. 6, with a similar budget of FLOPs/memory, $\epsilon$-only truncation achieves a lower accuracy compared to $\epsilon$ truncation with oversampling.

**Efficient training-aware subspaces.** INSTANT allows for selecting adaptive subspaces customized for each layer, via truncation strategies (Sec. 3.2). With different datasets, to keep the same amount of information, there is a large difference in the subspace ranks, which makes FLOPs vary during the training process (Appendix K.6). This rank adaptiveness controls the loss of tensor information,

Table 4: Ablation study of partial compressions, across 5 different CV datasets when fine-tuning the last block of EfficientFormer-L1. We report the MFLOPs and memory (Mem) required for training a single sample.

| Method | p | MFLOPs ↓ | Mem (MB) ↓ | Datasets | | | | | mAcc ↑ |
|---|---|---|---|---|---|---|---|---|---|
| | | | | CF100 ↑ | CF10 ↑ | Flowers ↑ | Food ↑ | Pets ↑ | |
| Vanilla | – | 1484 | 1.95 | 79.28 | 95.23 | 95.5 | 84.04 | 93.13 | 89.44 |
| INSTANT (compress $\mathbf{g_y}$) | 0 | 973 | 1.95 | 78.45 | 95.02 | 95.74 | 82.91 | 93.08 | 89.04 |
| | 5 | 1069 | 1.95 | 79.00 | 95.26 | 96.16 | 83.49 | 93.02 | 89.39 |
| | 7 | 1104 | 1.95 | 79.18 | 95.39 | 95.95 | 83.73 | 93.08 | 89.47 |
| INSTANT (compress $\mathbf{x}$) | 0 | 796 | 0.12 | 78.80 | 95.07 | 94.99 | 82.63 | 92.83 | 88.86 |
| | 5 | 885 | 0.33 | 78.99 | 95.35 | 95.37 | 83.01 | 92.75 | 89.09 |
| | 7 | 921 | 0.41 | 78.93 | 95.40 | 95.43 | 83.19 | 92.80 | 89.15 |
| INSTANT | 0 | 270 | 0.16 | 77.64 | 94.66 | 92.23 | 81.97 | 92.64 | 87.83 |
| | 5 | 475 | 0.38 | 78.65 | 95.07 | 95.93 | 82.84 | 93.21 | 89.14 |
| | 7 | 544 | 0.45 | 79.01 | 95.23 | 95.92 | 83.05 | 93.02 | 89.25 |

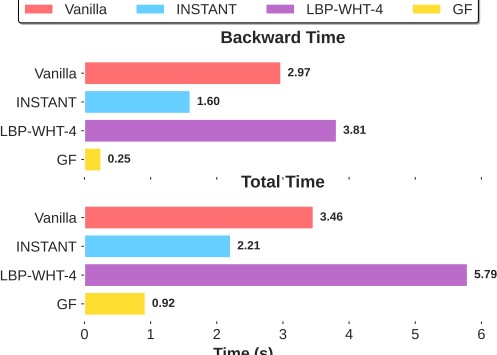

Figure 5: Training time of EfficientFormer-L1 on CIFAR10 on a Raspberry Pi 5.

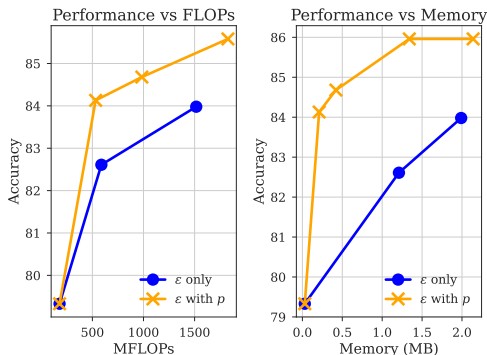

Figure 6: INSTANT performance with oversampling

leading to a small drop in accuracy while reducing a huge number of FLOPs by eliminating the projection onto redundant subspaces. In addition, this also helps to achieve more efficient memory storage. As we observe, with the same energy threshold $\epsilon$, the rank $R_x$ of truncated activation $\hat{\mathbf{x}}$ is typically smaller than the rank $R_y$ of truncated gradients $\hat{\mathbf{g}}_\mathbf{y}$ and much smaller than the natural rank $L$ of activation $\mathbf{x}$. Therefore, INSTANT requires much lower activation storage than Vanilla.

## 5 CONCLUSION

In this study, we introduced INSTANT, a highly efficient backpropagation method for Transformer models that targets two key bottlenecks in resource-efficient training: memory consumption and computational complexity. INSTANT constructs efficient low-rank projectors to compress both the activation during the forward pass and the gradient of the output during the backward pass (Sec. 3). We demonstrate the low-rank nature of the activation gradient (Sec. 4.2), and show that jointly compressing activation and gradient substantially reduces the computational overhead of the training (Sec. 4.3). This work investigates the relatively underexplored area of gradient compression and low-cost tensor decomposition as a means of scalable, resource-efficient model training.

### ACKNOWLEDGEMENTS

Part of this work was funded by Hi!PARIS Center on Data Analytics and Artificial Intelligence, by the European Union's HORIZON Research and Innovation Programme under grant agreement No 101120657, project ENFIELD (European Lighthouse to Manifest Trustworthy and Green AI) and by

French National Research Agency (ANR-22-PEFT-0003 and ANR-22-PEFT-0007) as part of France 2030, the NF-NAI project and NF-FITNESS project.

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

# A  MATH NOTATIONS.

We provide the table of all our math notations:

Table 5: Table of symbols

| Symbol | Space | Meaning |
|--------|-------|---------|
| $L$ | $\mathbb{R}$ | Sequence length |
| $C_x$ | $\mathbb{R}$ | Layer input features |
| $C_y$ | $\mathbb{R}$ | Layer output features |
| $R_x$ | $\mathbb{R}$ | Rank of activation in low-rank space |
| $R_y$ | $\mathbb{R}$ | Rank of gradient activation in low-rank space |
| $N_t$ | $\mathbb{R}$ | The number of training steps for each calibration |
| $\sigma_i$ | $\mathbb{R}$ | The $i$-th eigenvalue |
| $\epsilon$ | $\mathbb{R}$ | The energy threshold |
| $\mathcal{E}$ | $\mathbb{R}$ | Sum of squares of all eigenvalues |
| $k$ | $\mathbb{R}$ | The eigenvalue's index to keep $\epsilon$ total energy |
| $p$ | $\mathbb{R}$ | Oversampling value |
| $\mathcal{L}$ | $\mathbb{R}$ | Loss value used to backpropagate |
| $\mathbf{x}$ | $\mathbb{R}^{L \times C_x}$ | Input of a linear layer without batch dimension |
| $\mathbf{y}$ | $\mathbb{R}^{L \times C_y}$ | Output of a linear layer without batch dimension |
| $\mathbf{w}$ | $\mathbb{R}^{C_y \times C_x}$ | Weight of a linear layer |
| $\mathbf{g_x}$ | $\mathbb{R}^{L \times C_x}$ | Gradient w.r.t input of a linear layer without batch dimension |
| $\mathbf{g_y}$ | $\mathbb{R}^{L \times C_y}$ | Gradient w.r.t output of a linear layer without batch dimension |
| $\mathbf{P}$ | $\mathbb{R}^{R_x \times L}$ | Low-rank matrix to project activation into low-rank space |
| $\mathbf{Q}$ | $\mathbb{R}^{R_y \times L}$ | Low-rank matrix to project gradient activation into low-rank space |

# B  PROJECTION CONSTRUCTION ISSUES

This appendix provides details for Sec. 3.2, including explanation (Sec. $B$.1), theoretical foundation (Sec. $B$.2), and our efficient calibration approach (Sec. $B$.3).

## B.1  WHY ARE TWO COMPRESSION TENSORS REQUIRED FOR ACTIVATION AND GRADIENT COMPRESSION?

Low-rank approximation method reduces computational complexity by doing multiplications in the smaller low-rank space. Given $\mathbf{A} \cdot \mathbf{B}$ with $\mathbf{A} \in \mathbb{R}^{m \times n}$, $\mathbf{B} \in \mathbb{R}^{n \times p}$, this multiplication is of $\mathcal{O}(m \cdot n \cdot p)$ complexity. Low-rank approximation introduces low-rank matrix $\mathbf{P} \in \mathbb{R}^{r \times m}$ ($r \ll \min(m, n, p)$) such that $\mathbf{A} \approx \mathbf{P}^\top \cdot \mathbf{P} \cdot \mathbf{A}$. The original multiplication becomes:

$$\mathbf{A} \cdot \mathbf{B} \approx \mathbf{P}^\top \cdot \mathbf{P} \cdot \mathbf{A} \cdot \mathbf{B} = \mathbf{P}^\top \cdot ((\mathbf{P} \cdot \mathbf{A}) \cdot \mathbf{B}). \tag{6}$$

The right expression in $equation$ 6 is of complexity $\mathcal{O}(r \cdot (n \cdot p + m \cdot n + m \cdot p)) \ll \mathcal{O}(m \cdot n \cdot p)$.

The problem of this decomposition is $\mathbf{P}^\top \cdot \mathbf{P} \neq \mathbb{I}_{\mathbf{m} \times \mathbf{m}}$ with $\mathbb{I}_{m \times m}$ be the identity matrix of size $(m \times m)$, because $rank(\mathbf{P}^\top \cdot \mathbf{P}) \leq rank(\mathbf{P}) \leq r \ll m = rank(\mathbb{I}_{m \times m})$. Singular Value Decomposition (SVD) is used to find a good approximation that, given a full-rank matrix $\mathbf{A}$, SVD decomposes $\mathbf{A} = \mathbf{U} \cdot \mathbf{\Sigma} \cdot \mathbf{V}^\top$ with $\mathbf{U} \in \mathbb{R}^{\mathbf{m} \times \mathbf{m}}$ being left singular vectors. If we assign $\mathbf{P} = \mathbf{U}$ for multiplication in $equation$ 6, $\mathbf{A} = \mathbf{P}^\top \cdot \mathbf{P} \cdot \mathbf{A}$ because $\mathbf{P}^\top \cdot \mathbf{P} = \mathbb{I}_{\mathbf{m} \times \mathbf{m}}$, but the complexity of the approximation in $equation$ 6 is $\mathcal{O}(m \cdot (n \cdot p + m \cdot n + m \cdot p)) \geq \mathcal{O}(m \cdot n \cdot p)$.

Therefore, we need to truncate $\mathbf{U}$ to $\mathbf{U_r}$ (Sec. 3.2) to have $r \ll \min(m, n, q)$ to reduce computations. Although $\mathbf{U_r}^\top \cdot \mathbf{U_r} \neq \mathbb{I}_{\mathbf{m} \times \mathbf{m}}$, with appropriate truncation strategy, $\mathbf{U_r}^\top \cdot \mathbf{U_r} \cdot \mathbf{A} \approx \mathbf{A}$. This approximation with complexity of $\mathcal{O}(r \cdot (n \cdot p + m \cdot n + m \cdot p)) << \mathcal{O}(m \cdot n \cdot p)$ so assigning $\mathbf{P} = \mathbf{U_r}$ satisfies 6 and significantly reduces computational cost.

**Why do we need both $\mathbf{P}, \mathbf{Q}$ compression matrices?**. Why do we not only involve $\mathbf{P}$ for both activation compression($\mathbf{x} \xrightarrow{\mathbf{P}} \hat{\mathbf{x}}$) and gradient compression($\mathbf{g_y} \xrightarrow{\mathbf{P}} \hat{\mathbf{g}}_{\mathbf{y}}$)? Assuming that we decompose activation $\mathbf{x}$ and truncate its singular vectors to get $\mathbf{P} = \mathbf{U_r}$. The truncated $\mathbf{U_r}$ successfully

recovers $\mathbf{x}$ but does not work with $\mathbf{g_y}$ ($\mathbf{U_r^\top} \cdot \mathbf{U_r} \cdot \mathbf{g_y} \neq \mathbf{g_y}$) and $\mathbf{U_r^\top} \cdot \mathbf{U_r} \neq \mathbb{I}_{\mathbf{m \times m}}$ as well. Therefore, it is required to decompose $\mathbf{g_y}$ to get an additional compression matrix $\mathbf{Q}$ for a correct approximation of this output gradient.

## B.2 PROOF OF ESPACE THEOREM

Given vector $\mathbf{x} \in \mathbf{X}$, subspace $\mathbf{P}$ and $\tilde{\mathbf{x}} \in \tilde{\mathbf{X}}$ is the recovery of $\mathbf{x}$, the mean squared error (MSE) of the decomposition is defined as:

$$\mathbf{MSE}(\mathbf{x}, \tilde{\mathbf{x}}) = \mathbb{E}\|\mathbf{x} - \tilde{\mathbf{x}}\|^2, \tag{7}$$

where $\tilde{\mathbf{x}} = \mathbf{P}^\top \cdot \mathbf{P} \cdot \mathbf{x} = \sum_{i=1}^{L} \langle \mathbf{p_i}, \mathbf{x} \rangle \mathbf{p_i}$.

$\{\mathbf{p_i}\}_{i=1}^{L}$ are transpose of orthonormal row vectors of $\mathbf{P}$, i.e.,

$$\langle \mathbf{p_i}, \mathbf{p_j} \rangle = \mathbf{1}_{\{i=j\}}, \quad \forall i, j \in \{1, \dots, L\}. \tag{8}$$

Given $\mathbf{x}$ and $\tilde{\mathbf{x}}$, we examine the squared error:

$$\|\mathbf{x} - \tilde{\mathbf{x}}\|^2 = \|\mathbf{x}\|^2 + \|\tilde{\mathbf{x}}\|^2 - 2\mathbf{x}^\top \cdot \tilde{\mathbf{x}} \tag{9}$$

$$= \|\mathbf{x}\|^2 + \|\tilde{\mathbf{x}}\|^2 - 2\sum_{i=1}^{L}(\mathbf{p}_i^\top \cdot \mathbf{x}) \cdot \mathbf{p}_i^\top \cdot \mathbf{x} \tag{10}$$

$$= \|\mathbf{x}\|^2 + \|\tilde{\mathbf{x}}\|^2 - 2\sum_{i=1}^{L}(\mathbf{p}_i^\top \cdot \mathbf{x})^2. \tag{11}$$

We have $L_2 - norm$ of $\tilde{\mathbf{x}}$:

$$\|\tilde{\mathbf{x}}\|^2 = \tilde{\mathbf{x}}^\top \cdot \tilde{\mathbf{x}} = \left(\sum_{i=1}^{L}(\mathbf{p}_i^\top \cdot \mathbf{x}) \cdot \mathbf{p}_i\right)^\top \left(\sum_{i=1}^{L}(\mathbf{p}_i^\top \cdot \mathbf{x}) \cdot \mathbf{p}_i\right), \tag{12}$$

and since Eq. 8, we have:

$$\|\tilde{\mathbf{x}}\|^2 = \sum_{i=1}^{L}(\mathbf{p}_i^\top \cdot \mathbf{x})^2. \tag{13}$$

We reconsider the expression for the squared error:

$$\|\mathbf{x} - \tilde{\mathbf{x}}\|^2 = \|\mathbf{x}\|^2 + \sum_{i=1}^{L}(\mathbf{p}_i^\top \cdot \mathbf{x})^2 - 2\sum_{i=1}^{L}(\mathbf{p}_i^\top \cdot \mathbf{x})^2 \tag{14}$$

$$= \|\mathbf{x}\|^2 - \sum_{i=1}^{L}(\mathbf{p}_i^\top \cdot \mathbf{x})^2. \tag{15}$$

We take expectation on both sides and obtain a formula for the MSE:

$$\mathbb{E}\left[\|\mathbf{x} - \tilde{\mathbf{x}}\|^2\right] = \mathbb{E}\left[\|\mathbf{x}\|^2\right] - \mathbb{E}\left[\sum_{i=1}^{L}(\mathbf{p}_i^\top \cdot \mathbf{x})^2\right] \tag{16}$$

$$= \mathbb{E}\left[\|\mathbf{x}\|^2\right] - \mathbb{E}\left[\sum_{i=1}^{L}\mathbf{p}_i^\top \cdot (\mathbf{x} \cdot \mathbf{x}^\top) \cdot \mathbf{p}_i\right] \quad (\mathbf{p_i^\top} \cdot \mathbf{x} = \mathbf{x}^\top \cdot \mathbf{p_i}) \tag{17}$$

$$= \mathbb{E}\left[\|\mathbf{x}\|^2\right] - \sum_{i=1}^{L}\mathbf{p}_i^\top \cdot \mathbb{E}\left[\mathbf{x} \cdot \mathbf{x}^\top\right] \cdot \mathbf{p}_i. \tag{18}$$

In this expression, $\mathbb{E}\left[\|\mathbf{x}\|^2\right]$ does not depend on $\{\mathbf{p}_i\}_{i=1}^{L}$, and therefore, minimizing the MSE is equivalent to **maximizing** the following expression:

$$\sum_{i=1}^{L}\mathbf{p}_i^\top \cdot \mathbb{E}\left[\mathbf{x} \cdot \mathbf{x}^\top\right] \cdot \mathbf{p}_i = \sum_{i=1}^{L}\mathbf{p}_i^\top \cdot \mathbf{C_x} \cdot \mathbf{p}_i. \tag{19}$$

Because the set $\{\mathbf{p}_i\}_{i=1}^L$ forms an orthonormal basis, the expression can be interpreted as sum of Rayleigh quotients: $\sum_{i=1}^L \mathcal{R}(\mathbf{C_x}, \mathbf{p_i}) = \sum_{i=1}^L \mathbf{p}_i^\top \cdot \mathbf{C_x} \cdot \mathbf{p}_i$. The Rayleigh quotient is maximized when $\{\mathbf{p}_i\}_{i=1}^L$ are the eigenvectors corresponding to the largest eigenvalues of $\mathbf{C_x}$. Therefore, the optimal choice is to select $\{\mathbf{p}_i\}_{i=1}^L$ as the top $L$ eigenvectors of the auto-correlation $\mathbf{C_x}$.

### B.3 LOW-COST CALIBRATION PROOF

Considering our targets of reducing both peak memory and FLOPs, our calibration needs to be efficient, i.e does **not increase peak memory** and adds **small extra computation cost**.

**INSTANT calibration does not increase peak memory.** To reduce peak memory consumption in the calibration step, we propose 2 strategies:

- **On-policy processing.** Instead of saving multiple $\mathbf{x}$ and $\mathbf{g_y}$ for post-processing, after each calibrating iteration, the data batch is processed to store key elements and deleted afterwards. This approach does not increase the processing time.

- **Smaller batch.** In each training step, $B$ samples are put into the system. In calibration, we set a batch to $\frac{B}{n_s}$ samples, where $n_s$ is the number of mini-batches inside one batch.

---

**Algorithm 1** Efficient Calibration Without Memory Accumulation

---

**Require:** Model $\mathcal{M}$, number of calibration iterations $N_c$, calibration batch samples $B_c = \frac{B}{n_s}$, initial parameters $\theta_0$.

1: **Data processing** Calibration dataloader $D_c$ containing $N_c \cdot B_c$ calibration samples.

2: **Initializing** Set 2 dictionaries $A$ and $G$

3: **for** each calibration batch $b_c \in D_c$ **do**

4:   Freeze model parameters and optimizer states.

5:   **for** each layer $l$ **do**

6:     $\mathbf{y}, \mathbf{x}, \mathbf{g}_y \leftarrow \mathcal{M}(b_c)$

7:     $\mathbf{C}_X \leftarrow \mathbf{x} \cdot \mathbf{x}^\top$ $\qquad\qquad\qquad\qquad\qquad\qquad\qquad\qquad \mathbf{C}_G \leftarrow \mathbf{g}_y \cdot \mathbf{g}_y^\top$

8:     $s_X \leftarrow \sum_{i=1}^{B_c} \mathbf{C}_X[i]$ $\qquad\qquad\qquad\qquad\qquad\qquad\quad s_G \leftarrow \sum_{i=1}^{B_c} \mathbf{C}_G[i]$

9:     $A[l] \leftarrow A[l] + s_X$ $\qquad\qquad\qquad\qquad\qquad\qquad\qquad\quad G[l] \leftarrow G[l] + s_G$

10:   **end for**

11: **end for**

12: **for** each layer $l$ **do**

13:   $U_X, \Sigma_X \leftarrow \text{SVD}(A[l])$ $\qquad\qquad\qquad\qquad\qquad\qquad\quad U_G, \Sigma_G \leftarrow \text{SVD}(G[l])$

14:   $P \leftarrow \text{OVERSAMPLE}(U_X^\top)$ $\qquad\qquad\qquad\qquad\qquad\quad Q \leftarrow \text{OVERSAMPLE}(U_G^\top)$

15: **end for**

---

Instead of storing $\mathbf{x}, \mathbf{g_y}$ with $4 \cdot N_L \cdot B \cdot L \cdot C_x$ and $4 \cdot N_L \cdot B \cdot L \cdot C_y$ bytes, respectively, we just store 2 dictionaries $A$, $G$. These dictionaries have the same size of $4 \cdot N_L \cdot L^2$ bytes (Step 9 in Algorithm 1) with $N_L$ as the number of layers that we implement INSTANT. In application, we adaptively calibrate to get each dictionary of size $4 \cdot N_L \cdot \min(L, C)^2$ and $N_L < B$ to preserve memory.

In summary, the calibration step does not increase peak memory due to on-policy processing of $\mathbf{x}, \mathbf{g_y}$, smaller batch, and freezing optimizer states strategies (Alg. 1). Moreover, we propose a dynamic data processing in which the number of mini-batches $n_s$ can be adaptively increased to reduce peak memory of the calibration step.

**INSTANT calibration adding small extra computation.** thanks to long-term subspaces utilization, which is proved via the following analysis of the ratio between calibration FLOP ($flop_c$) and training FLOP ($flop_t$).

- **Experiment setup:** 5 calibration iterations for each $N_t = 200$ training iteration (as in our experiments). For the sake of simplicity, our linear layer: $y = x \cdot w^T$ with $x \in \mathbb{R}^{B \times L \times C_x}, w \in \mathbb{R}^{C_y \times C_x}, y \in \mathbb{R}^{B \times L \times C_y}$.

- **FLOP training:** $flop_t = 200 \cdot (6 \cdot B \cdot L \cdot c_x \cdot C_y)$ (forward and backward pass)

- **FLOP calibration (Alg.1):** $f_c = 5 \cdot (6 \cdot B \cdot L \cdot C_x \cdot C_y)$ (step 6) $+ 5 \cdot (2 \cdot B \cdot L^2 \cdot (C_x + C_y))$ (step 7) $+ [4/3 \cdot L^3 + 4/3 \cdot L^3]$ (SVD cost, step 13).

- **Ratio:** $f_c/f_t = \frac{30 \cdot B \cdot L \cdot C_x \cdot C_y + 10 \cdot B \cdot L^2 \cdot (C_x + C_y) + 8/3 \cdot L^3)}{1200 \cdot B \cdot L \cdot C_x \cdot C_y} \quad \frac{1}{40}(L < min(C_x, C_y))$

Therefore, FLOP of calibration is small compared to FLOP of training. Taking Tab.1 as an example, INSTANT-5 with reported **475** MFLOPs for finetuning EfficientFormerL1 requires extra **37** MFLOPs for calibration. Therefore, only reporting training flop still reflects a fair comparison with other methods.

## C  PROPAGATED ERROR ANALYSIS

Similar to Sec. 3.2, we denote $\mathbb{E}\left[\mathbf{g_y} \cdot \mathbf{g_y}^\top\right] = \mathbf{C_G} = \sum_{i=1}^{L} \sigma_i \cdot \mathbf{u}_i \cdot \mathbf{u}_i^\top$, where $\mathbf{u}_i$ is the $i^{th}$ column of left singular vectors $\mathbf{Q}$ when decomposing $\mathbf{C_G}$. Let $\mathbf{M} = \mathbf{I} - \frac{\mathbf{Q}^\top \cdot \mathbf{Q}}{\epsilon}$, notably that $\mathbf{Q}^\top \cdot \mathbf{Q} = \sum_{i=1}^{R_y} \mathbf{u}_i \cdot \mathbf{u}_i^\top$. The signal-to-noise ratio is calculated as:

$$SNR = \frac{\mathbb{E}\left[\left\|\frac{\mathbf{Q}^\top \cdot \mathbf{Q}}{\epsilon} \cdot \mathbf{g_y} \cdot \mathbf{w}\right\|_2^2\right]}{\mathbb{E}\left[\left\|\left(\mathbf{I} - \frac{\mathbf{Q}^\top \cdot \mathbf{Q}}{\epsilon}\right) \cdot \mathbf{g_y} \cdot \mathbf{w}\right\|_2^2\right]}. \tag{20}$$

We have: $\mathbf{M} = \mathbf{I} - \frac{\mathbf{Q}^\top \cdot \mathbf{Q}}{\epsilon} = \sum_{i=1}^{R_y} \frac{\epsilon - 1}{\epsilon} \cdot \mathbf{u}_i \cdot \mathbf{u}_i^\top + \sum_{i=R_y+1}^{L} \mathbf{u}_i \cdot \mathbf{u}_i^\top$,

and: $\|\mathbf{M} \cdot \mathbf{g_y} \cdot \mathbf{w}\|_2^2 = \sum_{j=1}^{C_x} \|\mathbf{M} \cdot \mathbf{g_y} \cdot \mathbf{w}_j\|_2^2$, where:

$$\mathbf{M} \cdot \mathbf{g_y} \cdot \mathbf{w}_j = \sum_{i=1}^{R_y} \frac{\epsilon - 1}{\epsilon} \left(\mathbf{u}_i \cdot \left(\mathbf{u}_i^\top \cdot \mathbf{g_y} \cdot \mathbf{w}_j\right)\right) + \sum_{i=R_y+1}^{L} \left(\mathbf{u}_i \cdot \left(\mathbf{u}_i^\top \cdot \mathbf{g_y} \cdot \mathbf{w}_j\right)\right) \tag{21}$$

$$= \sum_{i=1}^{R_y} \frac{\epsilon - 1}{\epsilon} \left(\mathbf{u}_i^\top \cdot \mathbf{g_y} \cdot \mathbf{w}_j \cdot \mathbf{u}_i\right) + \sum_{i=R_y+1}^{L} \left(\mathbf{u}_i^\top \cdot \mathbf{g_y} \cdot \mathbf{w}_j \cdot \mathbf{u}_i\right). \tag{22}$$

As a result:

$$\|\mathbf{M} \cdot \mathbf{g}_y \cdot \mathbf{w}_j\|_2^2 = \sum_{i=1}^{R_y} \left(\frac{\epsilon-1}{\epsilon}\right)^2 \left(\mathbf{u}_i^\top \cdot \mathbf{g}_y \cdot \mathbf{w}_j \cdot \mathbf{u}_i\right)^\top \cdot \left(\mathbf{u}_i^\top \cdot \mathbf{g}_y \cdot \mathbf{w}_j \cdot \mathbf{u}_i\right)$$
$$+ \sum_{i=R_y+1}^{L} \left(\mathbf{u}_i^\top \cdot \mathbf{g}_y \cdot \mathbf{w}_j \cdot \mathbf{u}_i\right)^\top \cdot \left(\mathbf{u}_i^\top \cdot \mathbf{g}_y \cdot \mathbf{w}_j \cdot \mathbf{u}_i\right) \tag{23}$$

$$= \sum_{i=1}^{R_y} \left(\frac{\epsilon-1}{\epsilon}\right)^2 \left(\mathbf{u}_i^\top \cdot \mathbf{g}_y \cdot \mathbf{w}_j\right) \cdot \mathbf{u}_i^\top \cdot \left(\mathbf{u}_i^\top \cdot \mathbf{g}_y \cdot \mathbf{w}_j\right) \cdot \mathbf{u}_i$$
$$+ \sum_{i=R_y+1}^{L} \left(\mathbf{u}_i^\top \cdot \mathbf{g}_y \cdot \mathbf{w}_j \cdot \mathbf{u}_i\right)^\top \cdot \left(\mathbf{u}_i^\top \cdot \mathbf{g}_y \cdot \mathbf{w}_j \cdot \mathbf{u}_i\right) \tag{24}$$

$$= \sum_{i=1}^{R_y} \left(\frac{\epsilon-1}{\epsilon}\right)^2 \left(\mathbf{w}_j^\top \cdot \mathbf{g}_y \cdot \mathbf{u}_i\right) \cdot \left(\mathbf{u}_i^\top \cdot \mathbf{g}_y \cdot \mathbf{w}_j\right) \cdot \mathbf{u}_i^\top \cdot \mathbf{u}_i$$
$$+ \sum_{i=R_y+1}^{L} \left(\mathbf{w}_j^\top \cdot \mathbf{g}_y \cdot \mathbf{u}_i\right) \cdot \left(\mathbf{u}_i^\top \cdot \mathbf{g}_y \cdot \mathbf{w}_j\right) \cdot \mathbf{u}_i^\top \cdot \mathbf{u}_i \tag{25}$$

$$= \sum_{i=1}^{R_y} \left(\frac{\epsilon-1}{\epsilon}\right)^2 \left(\mathbf{w}_j^\top \cdot \mathbf{g}_y \cdot \mathbf{u}_i\right) \cdot \left(\mathbf{u}_i^\top \cdot \mathbf{g}_y \cdot \mathbf{w}_j\right)$$
$$+ \sum_{i=R_y+1}^{L} \left(\mathbf{w}_j^\top \cdot \mathbf{g}_y \cdot \mathbf{u}_i\right) \cdot \left(\mathbf{u}_i^\top \cdot \mathbf{g}_y \cdot \mathbf{w}_j\right). \tag{26}$$

Therefore, the expectation is calculated as:

$$\mathbb{E}\left[\|\mathbf{M} \cdot \mathbf{g_y} \cdot \mathbf{w}\|_2^2\right] = \sum_{j=1}^{C_x} \left[\sum_{i=1}^{R_y} \left(\frac{\epsilon-1}{\epsilon}\right)^2 \mathbf{w}_j^\top \cdot \mathbb{E}\left[\mathbf{g_y}^\top \cdot \mathbf{u}_i \cdot \mathbf{u}_i^\top \cdot \mathbf{g_y}\right] \cdot \mathbf{w}_j\right]$$
$$+ \sum_{j=1}^{C_x} \left[\sum_{i=R_y+1}^{L} \mathbf{w}_j^\top \cdot \mathbb{E}\left[\mathbf{g_y}^\top \cdot \mathbf{u}_i \cdot \mathbf{u}_i^\top \cdot \mathbf{g_y}\right] \cdot \mathbf{w}_j\right] \tag{27}$$

$$= \sum_{j=1}^{C_x} \left[\sum_{i=1}^{R_y} \left(\frac{\epsilon-1}{\epsilon}\right)^2 \sigma_i \cdot \mathbf{w}_j^\top \cdot \mathbf{w}_j + \sum_{i=R_y+1}^{L} \sigma_i \cdot \mathbf{w}_j^\top \cdot \mathbf{w}_j\right] \tag{28}$$

$$= \left[\frac{(1-\epsilon)^2}{\epsilon^2} \sum_{i=1}^{R_y} \sigma_i + \sum_{i=R_y+1}^{L} \sigma_i\right] \cdot \|W\|_2^2. \tag{29}$$

Similarly, we have:

$$\mathbb{E}\left[\left\|\frac{\mathbf{Q}^\top \cdot \mathbf{Q}}{\epsilon} \cdot \mathbf{g_y} \cdot \mathbf{w}\right\|_2^2\right] = \frac{1}{\epsilon^2} \cdot \sum_{i=1}^{R_y} \sigma_i \cdot \|W\|_2^2. \tag{30}$$

Therefore, the signal-to-noise ratio is computed as:

$$SNR = \frac{\mathbb{E}\left[\left\|\frac{\mathbf{Q}^\top \cdot \mathbf{Q}}{\epsilon} \cdot \mathbf{g_y} \cdot \mathbf{w}\right\|_2^2\right]}{\mathbb{E}\left[\|\mathbf{M} \cdot \mathbf{g_y} \cdot \mathbf{w}\|_2^2\right]} = \frac{\frac{1}{\epsilon^2} \sum_{i=1}^{R_y} \sigma_i}{\frac{(1-\epsilon)^2}{\epsilon^2} \sum_{i=1}^{R_y} \sigma_i + \sum_{i=R_y+1}^{L} \sigma_i}. \tag{31}$$

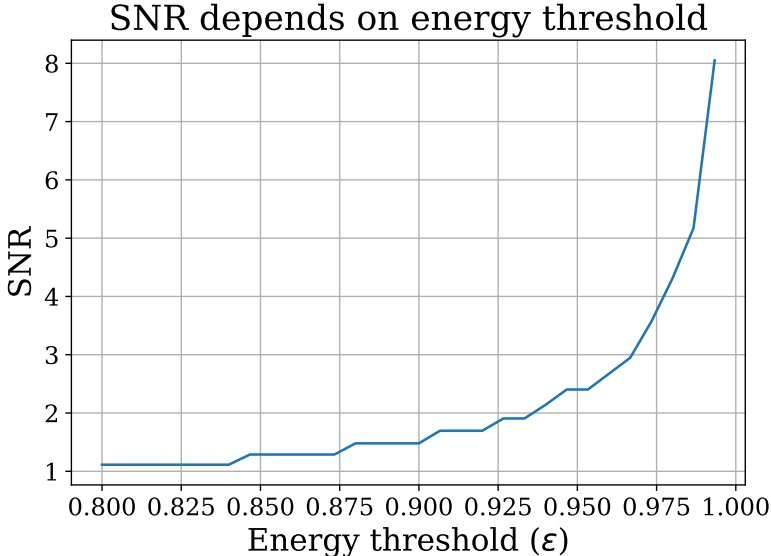

Figure 7: Signal-to-noise ratio analysis of one layer. When the energy threshold goes to 1, the SNR ratio will increase enormously. With a large enough value of energy threshold and oversampling, INSTANT can guarantee a high SNR ratio of training.

When $\epsilon \to 1$, $R_y \to L$, therefore, $SNR \to \infty$, and error will become 0 as shown in Fig. 7. Here, it is noticeable that the plot has some segments with breaks. This is because with some near $\epsilon$, the same rank $R_y$ is selected.

## D    FLOPs AND MEMORY ANALYSIS

In this section, the FLOPs and memory analysis for our INSTANT (Sec. D.1) and LBP-WHT (Sec. D.3) are described.

### D.1    FLOPs AND MEMORY ANALYSIS FOR INSTANT

By compressing activation at the forward pass, we are able to reduce the matrix storage from $\mathbb{R}^{L \times C_x} \to \mathbb{R}^{R_x \times C_x}$ by only storing two additional tensors $\mathbf{P} \in \mathbb{R}^{R_x \times L}$ and $\mathbf{Q} \in \mathbb{R}^{R_y \times L}$. Assuming that the activation is stored in fp32, which is 4 bytes for each value, the total memory for storing an activation sample at one layer reduces from $4 \cdot L \cdot C_x$ to $4 \cdot R_x \cdot C_x$. It is noticeable that $\mathbf{P}$ and $\mathbf{Q}$ remain constant across batches, meaning that even when training in a mini-batch setup, as typically do, $\mathbf{P}$ and $\mathbf{Q}$ are still two-dimensional tensors. This ensures that the memory cost for storing them is negligible.

In terms of computation, our training involves three main computational cost components: low-rank projection, low-rank matrix multiplication, and reverse projection. Since there are three low-rank projections, the total FLOPs of these projections $FLOP_p$ is:

$$FLOP_p = 2 \cdot R_x \cdot L \cdot C_x + 2 \cdot R_y \cdot L \cdot C_y + 2 \cdot R_x \cdot L \cdot C_y. \tag{32}$$

Eq. 4 and Eq. 5 illustrate two low-rank matrix multiplications of INSTANT. Since both of them are computed in low-rank spaces, the computational complexity $FLOP_m$ is:

$$FLOP_m = 2 \cdot R_y \cdot C_y \cdot C_x + 2 \cdot R_x \cdot C_x \cdot C_y. \tag{33}$$

The cost for reverse projection of $\widetilde{\mathbf{g}}_{\mathbf{x}}$ is $FLOP_r = 2 \cdot R_y \cdot L \cdot C_x$. Ultimately, the total computational complexity of our backpropagation process $FLOP_t$ is:

$$FLOP_t = FLOP_p + FLOP_m + FLOP_r = 2(R_x + R_y)(C_x \cdot C_y + L \cdot C_x + L \cdot C_y) \tag{34}$$

Note that this is the computational cost of one sample at one layer; for training one batch of $B$ samples, the total computational cost is $B \cdot FLOP_t$.

## D.2 FLOPs and memory analysis for Gradient Filter

The Gradient Filter splits the gradient and activations into small patches, then applies pooling to each patch to reduce the number of unique elements. By storing only these unique elements in the memory, the Gradient Filter can save $64\times$ memory compared to vanilla training with a patch size $8 \times 8$. Let:

$$p_h = \lceil h/8 \rceil, \qquad p_w = \lceil w/8 \rceil. \tag{35}$$

Gradient Filter compresses the activation $\mathbf{x} \in \mathbb{R}^{L \times C_x}$, with $L = h \times w$ into $\hat{\mathbf{x}} \in \mathbb{R}^{L' \times C_x}$, with $L' = p_h \times p_w$. The activation storage is calculated as $4 \cdot C_x \cdot p_h \cdot p_w$. Similarly, the gradient $\mathbf{g_y} \in \mathbb{R}^{L \times C_y}$ is compressed into $\hat{\mathbf{g}}_\mathbf{y} \in \mathbb{R}^{L' \times C_y}$. The total compression cost for activation and gradient is:

$$FLOP_c = (C_x + C_y) \times p_h \times p_w \times 8^2 \tag{36}$$

After compressing, by doing both 2 matrix multiplications in the low-rank space, the total computational cost in backpropagation of Gradient Filter is:

$$FLOP_b = 4 \times C_x \times C_y \times L' = 4 \times C_x \times C_y \times p_h \times p_w \tag{37}$$

Finally, the total computational cost of backpropagation with Gradient Filter is:

$$FLOP = FLOP_b + FLOP_c = 4 \times C_x \times C_y \times p_h \times p_w + (C_x + C_y) \times p_h \times p_w \times 8^2 \tag{38}$$

## D.3 FLOPs and memory analysis for LBP-WHT

LBP-WHT is originally designed for computer vision tasks. Considering a linear layer, let $\mathbf{x} \in R^{L \times C_x}$ as the input. LBP-WHT first splits the sequence length into $H$ and $W$, such that $L = H.W$, then reshapes to $\mathbf{x} \in \mathbb{R}^{C_x \times H \times W}$. Afterward, at each channel, the matrix shape $[H, W]$ is split into ($p_h \cdot p_w$) patches ($8 \times 8$) with:

$$p_h = \lceil h/8 \rceil, \qquad p_w = \lceil w/8 \rceil. \tag{39}$$

At each patch, the Walsh-Hadamard transformation is applied with only $R$ out of the total 64 bases of the WHT, where $R$ is the hyperparameter. The FLOPs of this projection $WFLOP_{px}$ is :

$$WFLOP_{px} = 2 \cdot R \cdot C_x \cdot p_h \cdot p_w \cdot 8^2, \tag{40}$$

where $8^2$ is because of the patch size $8 \times 8$. This compressed version is stored in memory instead of the original $\mathbf{x}$, therefore, the activation storage is: $4 \cdot R \cdot C_x \cdot p_h \cdot p_w$.

In the backward pass, the gradient is projected by the same transformation as the activation. Therefore, the computational cost $WFLOP_{pg}$ is:

$$WFLOP_{pg} = 2 \cdot R \cdot C_y \cdot p_h \cdot p_w \cdot 8^2. \tag{41}$$

LBP-WHT does two matrix multiplications in the low-rank space to calculate the weight gradient and compressed input gradient. The total cost of these multiplications is:

$$WFLOP_m = 2 \cdot R \cdot C_y \cdot p_h \cdot p_w \cdot C_x + 2 \cdot R \cdot C_y \cdot C_x \cdot p_h \cdot p_w + R \cdot C_x \cdot C_y. \tag{42}$$

Finally, the compressed input gradient is mapped to the original space to propagate to the previous layer with cost:

$$WFLOP_r = 2 \cdot R \cdot C_x \cdot p_h \cdot p_w \cdot 8^2. \tag{43}$$

In conclusion, the total FLOPs of LBP-WHT is computed as:

$$WFLOP = WFLOP_{px} + WFLOP_{pg} + WFLOP_m + WFLOP_r \tag{44}$$
$$= 4 \cdot R \cdot C_x \cdot C_y \cdot p_h \cdot p_w + R \cdot C_x \cdot C_y + 256 \cdot R \cdot C_x \cdot p_h \cdot p_w + 128 \cdot R \cdot C_y \cdot p_h \cdot p_w \tag{45}$$

We recommend that the readers be familiar with LBP-WHT (Yang et al., 2023b) paper and the source code for a better understanding of the calculation in each step.

# E  DATASET DESCRIPTION

**Computer vision tasks.**

- **CIFAR10** (CF10) (Krizhevsky et al., 2009) is a widely used image classification dataset consisting of 60,000 $32 \times 32$ color images across 10 classes, such as airplane, cat, and truck. It includes 50,000 training images and 10,000 test images, evenly distributed across classes.

- **CIFAR100** (CF100) (Krizhevsky et al., 2009) is an image classification dataset similar to CIFAR-10 but with 100 classes. It contains 60,000 color images of size $32 \times 32$ pixels—50,000 for training and 10,000 for testing. CIFAR-100 is more challenging than CIFAR-10 due to the larger number of categories and finer-grained distinctions between them, making it a popular benchmark for evaluating model performance on more complex classification tasks.

- **Flowers** (Nilsback & Zisserman, 2006) contains 8,189 images of 102 flower categories found in the UK. Each class has between 40 and 258 images. The images have high variability in scale, pose, and lighting, making the dataset a useful benchmark for testing the robustness of computer vision models in fine-grained classification.

- **Food-101** (Bossard et al., 2014) is a popular benchmark for food image classification, containing 101,000 images across 101 food categories such as pizza, sushi, and apple pie. Each class has 1,000 images—750 for training and 250 for testing—with real-world variability in presentation, lighting, and background.

- **The Oxford-IIIT Pet** (Parkhi et al., 2012) is a popular image dataset for pet classification and segmentation tasks. It contains 7,349 images of 37 different breeds of cats and dogs, with roughly 200 images per breed. The dataset features high variability in poses, lighting, and background, making it ideal for training and evaluating models in fine-grained classification.

**Natural language processing tasks.**

- **QNLI** (Question Natural Language Inference) has around 110k samples derived from the SQuAD dataset and reframed as a binary classification task, where the model determines if a given sentence contains the answer to a question (entailment or not).

- **SST-2** (Stanford Sentiment Treebank) is a sentiment analysis dataset with about 70k movie review sentences. Each sample is labeled as either positive or negative, making it a simple and clean binary classification task for sentiment understanding.

- **MNLI** (Multi-Genre Natural Language Inference) contains 433k sentence pairs from various text genres. The task is to determine whether a hypothesis is entailed, neutral, or contradicted by a given premise, making it a 3-class classification problem and a robust test of a model's reasoning ability.

- **CoLA** (Corpus of Linguistic Acceptability) comprises over 10k English sentences labeled as grammatically acceptable or unacceptable. It evaluates a model's ability to judge grammatical correctness, testing syntactic understanding.

- **MRPC** (Microsoft Research Paraphrase Corpus) includes around 5.8k sentence pairs from news sources, with binary labels indicating whether the two sentences are paraphrases. It's a common benchmark for paraphrase detection.

- **RTE** (Recognizing Textual Entailment) is a binary classification task with around 2.8k samples, where the goal is to decide whether a premise sentence entails a given hypothesis. It's a small dataset, but it's useful for testing generalization.

# F  THE REASON BEHIND OVERSAMPLING

In this section, we will prove our argument that oversampling is helpful since the energy threshold can only guarantee the amount of information when the calibration happens. More concretely, we will show that if we calibrate more, the oversampling is unnecessary. Conversely, we also show that oversampling can help even without calibration.

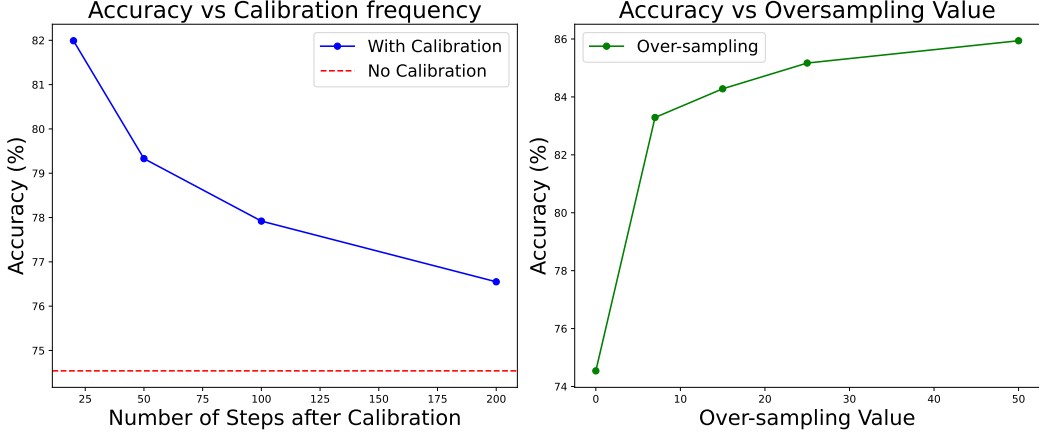

Figure 8: The effectiveness of oversampling. This experiment is conducted with BERT on the QNLI dataset.

Table 6: INSTANT can combine with LoRA to achieve a low-rank space of weight, activation, and gradient.

| Model | Method | MFLOPs $\downarrow$ | Mem $\downarrow$ | CF10 $\uparrow$ | CF100 $\uparrow$ |
|---|---|---|---|---|---|
| Efficient Former-L1 | LoRA | 896 | 1.95 | 94.48 | 77.27 |
| | LoRA + INSTANT-0 | 335 | 0.06 | 93.91 | 75.79 |
| | LoRA + INSTANT-5 | 413 | 0.26 | 94.35 | 76.73 |
| | LoRA + INSTANT-7 | 449 | 0.34 | 94.38 | 76,80 |

As shown in Fig. 8, without oversampling, increasing the calibration frequency (or reducing the number of steps after each calibration) results in a remarkable increase in accuracy. This shows that when the calibration happens more, the energy threshold is enough for capturing the information. However, since the cost for calibration is larger compared to training with INSTANT, oversampling can help in reducing the frequency of this phase. In addition, we experiment with only one calibration before training starts. Fig 8 (right) strengthens our argument by showing that without oversampling, the performance degrades significantly. Meanwhile, with a few values of oversampling, the accuracy increases approximately 10%.

## G  INSTANT IS ORTHOGONAL TO LoRA

LoRA (Hu et al., 2022), and its variances, are popular for low-rank adaptation. However, this research focuses on the low-rank characteristic of weights, while INSTANT is applied to the low-rank space of the activation and gradients. We, therefore, provide an additional ablation study to show that these 2 methods are orthogonal and can be combined.

While LoRA successfully reduces the number of trainable parameters, it still requires storing the original activations in memory to compute gradients, thus failing to reduce memory usage. Furthermore, LoRA necessitates the multiplication of two large matrices to compute the activation gradient, which is then backpropagated to the previous layer. As a result, the computational cost and memory of applying INSTANT are much smaller compared to LoRA only, as shown in Tab. 6.

## H  EXTENSION TO CONVOLUTION

### H.1  CONVERT CONVOLUTIONAL OPERATION TO LINEAR OPERATION

Many convolutional architectures (ResNet, VGGNet,...) utilize image-to-column transformation (Chellapilla et al., 2006) when working on convolutional operations. By stretching matrices,

this transformation turns a convolutional operation into a linear operation to utilize matrix multiplication optimization of deep learning libraries. We apply this idea to transform the convolutional operation into a linear operation, before applying tensor decomposition and low-rank backpropagation (Sec. 3.2, 3.3) to save memory and computation. Due to the low-cost transformation of image-to-column, we achieve up to 3x computation reduction.

## H.2 RESULTS

Fig. 9 shows the efficiency of INSTANT on 2 popular convolutional architectures: ResNet-50 and MobileNetV2, on 2 datasets CIFAR10 and CIFAR100. In all experiments, our method can save significant memory and FLOPs while maintaining high accuracy, compared to Vanilla. Specifically, INSTANT-5 and INSTANT-7 consistently outperform Vanilla in all three reported metrics. In ResNet-50, INSTANT-5 gains better performance with $3\times$ in FLOPs and $79\times$ in memory. These results indicate the efficiency of INSTANT on all architectures, including both Transformer-based and Convolutional-based models.

## I TRAINING LATENCY ANALYSIS

### I.1 EDGE DEVICES TRAINING LATENCY

Table 7: Iteration and backward time when finetuning 9 last layers of ViT-B/32 on CIFAR10. We also report backward speedup compared to Vanilla.

| Method | Raspberry Pi 5 | | | Intel E5-2267 | | |
|---|---|---|---|---|---|---|
| | Total time (s) | Backward time (s) | Speedup | Total time (s) | Backward time (s) | Speedup |
| Vanilla | 6.47 | 5.52 | $1\times$ | 2.69 | 1.80 | $1\times$ |
| Gradient Filtering | 1.85 | 0.84 | $2.17\times$ | 1.66 | 0.76 | $2.37\times$ |
| LBP-WHT-4 | 20.67 | 14.23 | $0.39\times$ | 14.83 | 9.76 | $0.18\times$ |
| INSTANT-0 | **1.43** | **0.44** | **12.55**$\times$ | **1.19** | **0.22** | **8.18**$\times$ |

We provide extra results of INSTANT on EfficientFormer-L1 on CPU Intel E5-2667 in Fig. 10. We conduct experiments with the same setup as in Sec. 4.1, using the PyTorch framework. Considering the larger architecture ViT-B/32, in which INSTANT-0 can save $17\times$ computation compared to Vanilla.

We provide additional results of INSTANT on ViT-B/32 using a Raspberry Pi 5 and an Intel E5-2267 CPU, as presented in Tab.7. Noticeably, we can save $8\times$ backward time on Intel E5-2267 and $12\times$ backward time on Raspberry Pi 5. The ($12\times$) time reduction is not comparable to ($17\times$) FLOP reduction (Tab.11). This reduction gap is reduced when implementing in other frameworks, such as CUDNN, as shown in Gradient Filter (Yang et al., 2023c). We also observe that the time reduction is strongly affected by the use of the device. However, these engineering-level optimizations fall outside the paper's scope. Generally, **computation reduction is converted into time reduction**, which proves the efficiency and applicability of INSTANT.

### I.2 GPU TRAINING LATENCY

We conducted experiments on the framework of MMCV, model EfficientFormerL1, with datasets CIFAR10, and with all methods (LBP-WHT, Gradient Filtering, INSTANT (ours), Vanilla) as provided in Tab. 1. We observed similar results on CIFAR100. As shown in Tab. 8, on V100 GPU, INSTANT's backward time is $1.4\times$ compared to Vanilla backwards time. The total training time (epoch time) of INSTANT is slightly higher than Vanilla. In short, INSTANT's FLOP reductions are not converted into time reduction like training on CPU.

## J ABLATION STUDY

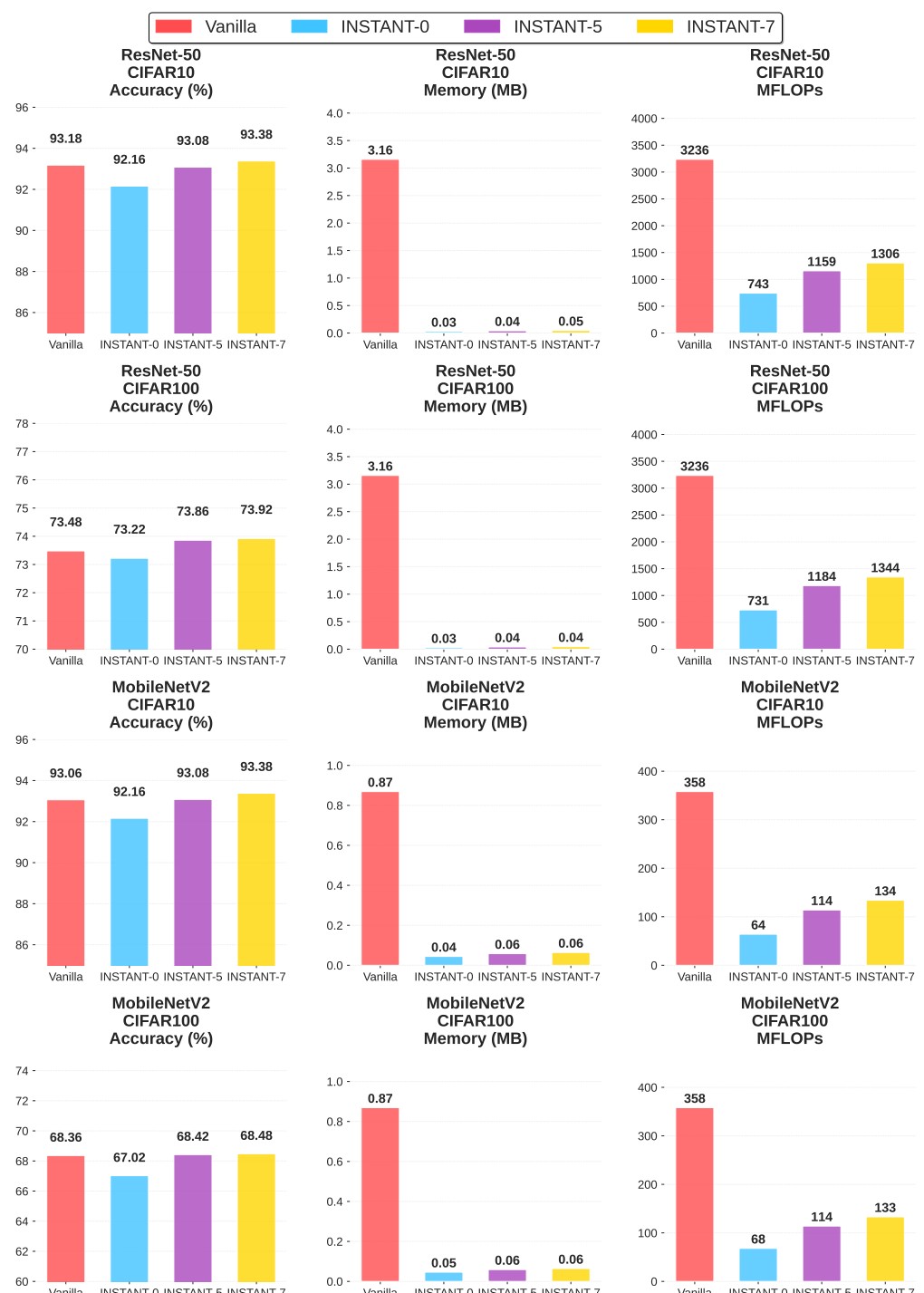

Figure 9: Performance of INSTANT on convolutional architectures.

## J.1 INSTANT WITHOUT CALIBRATION

To highlight the necessity and efficiency of our calibration scheme, we also provide extra baseline: Random - which is INSTANT using random subspaces $\mathbf{P}, \mathbf{Q}$ for compressing activation $\mathbf{x}$ and activation gradient $\mathbf{g_y}$. The difference is that in this Random scheme, we periodically update $\mathbf{P}, \mathbf{Q}$ by random Gaussian matrices each $N_t = 200$ training iterations.

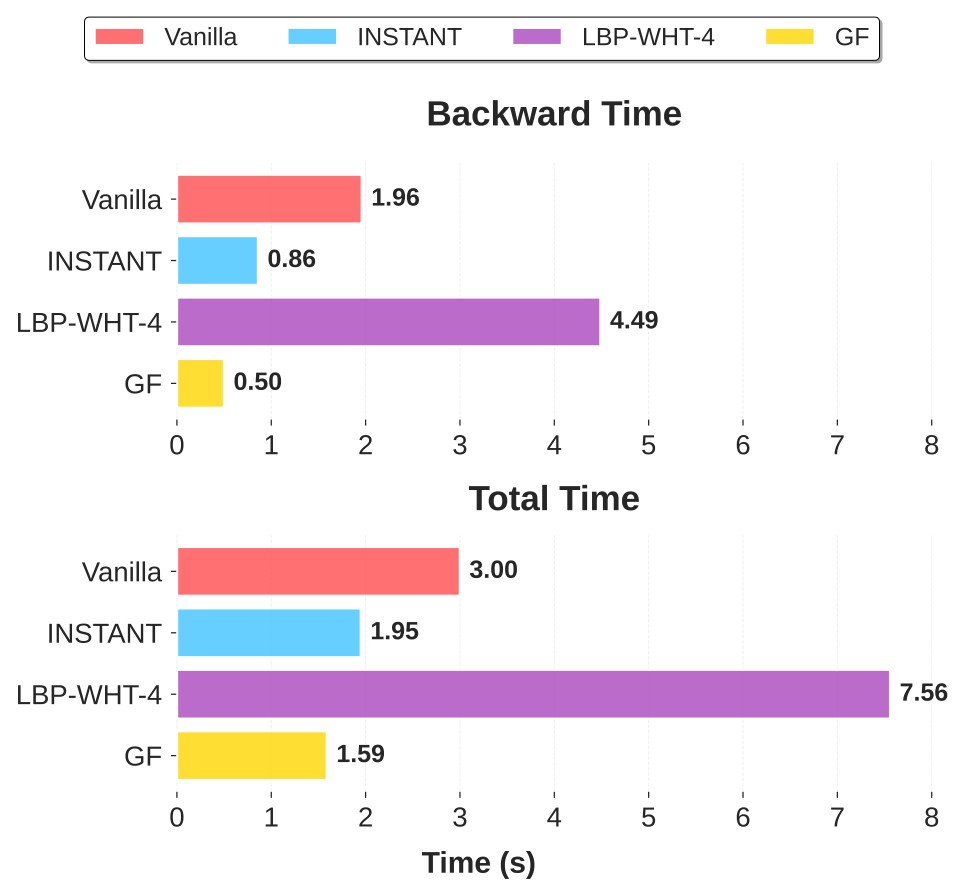

Figure 10: Training time over 1 epoch on CIFAR10 using a Intel E5-2667.

Table 8: Time reported for training 1 epoch with V100 GPU when partially and fully finetuning EfficientFormerL1 on CIFAR10. These reported numbers are averaged over 10 training epochs.

| Method / Time(s) | Partially Finetuning Last Layer | | | Full Finetuning | | |
|---|---|---|---|---|---|---|
| | Calibration* | Backward** | Total epoch*** | Calibration* | Backward** | Total epoch*** |
| Vanilla | 0 | 6.11 | 73.6 | 0 | 12.66 | 110.8 |
| Gradient Filtering | 0 | 7.77 | 74.0 | 0 | 15.07 | 94.2 |
| LBP-WHT-2 | 0 | 7.99 | 79.8 | 0 | 19.6 | 146.4 |
| LBP-WHT-4 | 0 | 9.02 | 123.8 | 0 | 18.55 | 246.0 |
| LBP-WHT-8 | 0 | 10.45 | 291.4 | 0 | 20.5 | 625.0 |
| INSTANT-0 | 2.16 | 8.8 | 74.2 | 4.99 | 17.55 | 116.6 |
| INSTANT-5 | 2.2 | 9.01 | 74.0 | 4.98 | 17.68 | 120.0 |
| INSTANT-7 | 2.22 | 8.73 | 74.2 | 4.98 | 17.17 | 120.2 |

(*) Calibration time is total time used for creating periodically updated subspaces during 1 training epoch
(**) Backward time is total time of `loss.backward()` during 1 training epoch.
(***) Epoch time is the total running time of training 1 epoch, including calibration time, forward pass, activation savings, loss calculation, backward pass, optimizer update,...

The experimental results (Tab. 9, Tab. 10) demonstrate that INSTANT significantly outperforms the Random projection method, where the compression rank is set to $\frac{N}{2}$ or $\frac{N}{4}$, with $N$ representing the input dimension. This highlights that INSTANT effectively requires a compact yet precise low-rank space to retain gradient information for accurate backpropagation through the network. Notably, in the case of full fine-tuning, the Random projection method fails to converge, as the gradients struggle to propagate through multiple layers, resulting in significant error accumulation due to the lack of a well-defined low-rank space.

Table 9: Partially finetuning last layer of EfficientFormer-L1 on CIFAR10, CIFAR100 with INSTANT-random (Appendix. J.1) and ESPACE (Appendix. K.1) methods.

| Method | CIFAR10 | CIFAR100 | Forward MFLOPs | Backward MFLOPs | Activation Mem (MB) |
|---|---|---|---|---|---|
| ESPACE – N/2 | 95.23 | 78.41 | 414 | 827 | 0.96 |
| ESPACE – N/4 | 94.21 | 76.26 | 212 | 414 | 0.48 |
| Random – N/2 | 86.52 | 73.13 | 742 | 827 | 0.96 |
| Random –N/4 | 64.97 | 46.02 | 742 | 414 | 0.48 |
| INSTANT-0 | 94.66 | 77.64 | 742 | 270 | 0.16 |
| INSTANT-5 | 95.07 | 78.65 | 742 | 475 | 0.38 |
| INSTANT-7 | 95.23 | 79.01 | 742 | 544 | 0.45 |
| Vanilla | 95.23 | 79.28 | 742 | 1484 | 1.95 |

Table 10: Full finetuning EfficientFormer-L1 on CIFAR10, CIFAR100 with INSTANT-random (Appendix. J.1) and ESPACE (Appendix. K.1) methods.

| Method | CIFAR10 | CIFAR100 | Forward MFLOPs | Backward MFLOPs | Activation Mem (MB) |
|---|---|---|---|---|---|
| ESPACE – N/2 | 93.16 | 74.56 | 2347 | 4694 | 4.29 |
| ESPACE – N/4 | 51.18 | 35.62 | 1173 | 2347 | 2.14 |
| Random – N/2 | 13.47 | 3.66 | 2264 | 4694 | 4.29 |
| Random – N/4 | 10.00 | 9.26 | 2264 | 2347 | 2.14 |
| INSTANT-5 | 96.29 | 82.41 | 2264 | 2107 | 1.98 |
| INSTANT-10 | *96.48* | *83.05* | 2264 | 2491 | 2.73 |
| INSTANT-15 | **96.85** | **83.56** | 2264 | 2884 | 3.45 |
| Vanilla | 96.99 | 84.84 | 2264 | 4528 | 18.46 |

## K  ADDITIONAL RESULTS

### K.1  COMPARISON OF INSTANT WITH ESPACE

ESPACE primarily focuses on compressing activations only, whereas our main objective is to jointly compress both activations and activation gradients. In trade-off to save forward computations, ESPACE performance is degraded compared to INSTANT due to error accumulations, even in the forward pass. Noticeably, using quite high dimension compressions $r = \frac{N}{2}$ or $r = \frac{N}{4}$, ESPACE observes limited overhead reductions compared to INSTANT. More experimental results are shown in Tab. 9 and Tab. 10.

### K.2  RESULTS ON LARGE ARCHITECTURE FOR VISION TASKS

Vision Transformer achieves extraordinary performance on image classification. We conduct experiments with ViT-B/32, a variant of Vision Transformer (Dosovitskiy et al., 2021) with 88 million parameters, which is **7**× EfficientFormer-L1 with 12 million parameters. In terms of accuracy, Gradient Filtering fails, while LBP-WHT-4 performs worse than EfficientFormer-L1, which suggests that their strategies are inappropriate for ViT-based architectures. Conversely, INSTANT with SVD-based compression can preserve significant information while achieving a good compression rate with negligible performance drop. Noticeably, INSTANT-7 outperforms Vanilla in all reported metrics, including accuracy.

### K.3  RESULTS ON COMPLEX PLACES-365 DATASET

We finetuned EfficientFormer-L1 on Places-365 (Zhou et al., 2017) dataset, which contains 1.8M images and is more challenging than ImageNet, i.e, models often perform worse on Places-365 than on the ImageNet dataset. The table above demonstrates that INSTANT remains highly effective even with large datasets such as Places-365. All three versions of INSTANT show only a minor reduction in performance compared to vanilla training, despite achieving a threefold reduction in

Table 11: Experimental results on CIFAR10 when fine-tuning 9 last layers of ViT-B/32. We report the MFLOPs and memory (Mem) required for training a single sample.

| Model | Method | MFLOPs ↓ | Mem ↓ | CF10 ↑ |
|---|---|---|---|---|
| ViT-B/32 | Vanilla | 2831 | 2.20 | 96.56 |
| | Gradient Filtering | 63 | 0.07 | 30.38 |
| | LBP-WHT-4 | 863 | 0.70 | 92.1 |
| | INSTANT-0 | 161 | 0.11 | 96.36 |
| | INSTANT-5 | 445 | 0.31 | 96.36 |
| | INSTANT-7 | 567 | 0.40 | 96.6 |

Table 12: We finetuned the last layer of EfficientFormer-L1 on Places-365 dataset.

| Method | Accuracy | Backward MFLOPs* | Activation Memory (MB) |
|---|---|---|---|
| Vanilla | 55.30 | 1484 | 1.95 |
| Gradient Filtering | 9.2 | 24 | 0.04 |
| LBP-WHT-2 | 50.92 | 95 | 0.12 |
| LBP-WHT-4 | 53.27 | 335 | 0.40 |
| LBP-WHT-8 | 54.67 | 1227 | 1.43 |
| INSTANT-0 | 54.32 | 388 | 0.37 |
| **INSTANT-5** | **54.57** | **567** | **0.60** |
| INSTANT-7 | 54.55 | 606 | 0.66 |

(*) The reported Backward MFLOPs includes forward compression in cases of INSTANT, Gradient Filtering, and LBP-WHT

both computational cost and memory usage. When compared to LBP-WHT, INSTANT offers slightly better performance while maintaining the same memory and computational budget. In contrast, Gradient Filtering fails to converge, yielding poor results.

### K.4 RESULTS ON LARGE LANGUAGE MODEL (TINYLLAMA)

We conduct some additional experiments on TinyLlama (Zhang et al., 2024) on the BoolQ (Clark et al., 2019) dataset with a similar setup to NLP tasks. The results are reported in Tab. 13. It is noticeable that, with a bigger model, INSTANT only requires a small rank to keep a large amount of energy, thus reducing a large amount of computational and memory consumption. However, in this experiment, although INSTANT witnesses a $3\%$ drop in performance compared to vanilla training, INSTANT can save about $13\times$ computational cost and $64\times$ memory consumption. This makes training a large model with more than a billion parameters on resource-constrained devices possible.

Table 13: TinyLlama results on BoolQ dataset

| Fine-tuning the Last Block | | | | |
|---|---|---|---|---|
| Model | Method | MFLOPs ↓ | Mem ↓ | BoolQ ↑ |
| TinyLLama | Vanilla | 90194 | 35 | 67.71 |
| | LBP-WHT-4 | 15083 | 5.47 | 63.94 |
| | INSTANT-0 | 5517 | 0.07 | 63.88 |
| | INSTANT-5 | 7298 | 0.41 | 64.77 |
| | INSTANT-7 | 7465 | 0.55 | 64.89 |

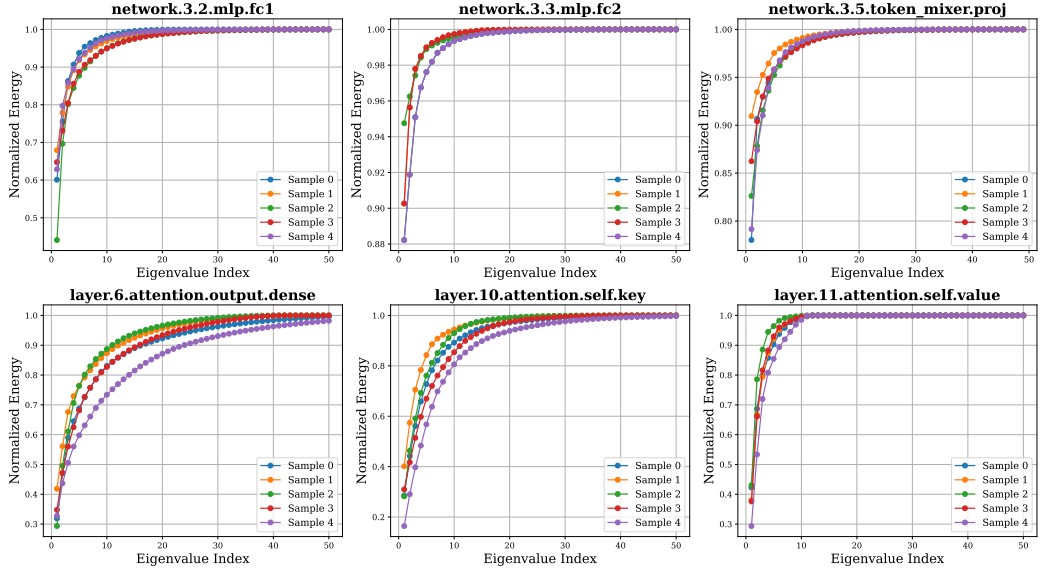

Figure 11: The normalized energy of eigenvalues of the gradient of many different layers in EfficientFormer-L1 and BERT.

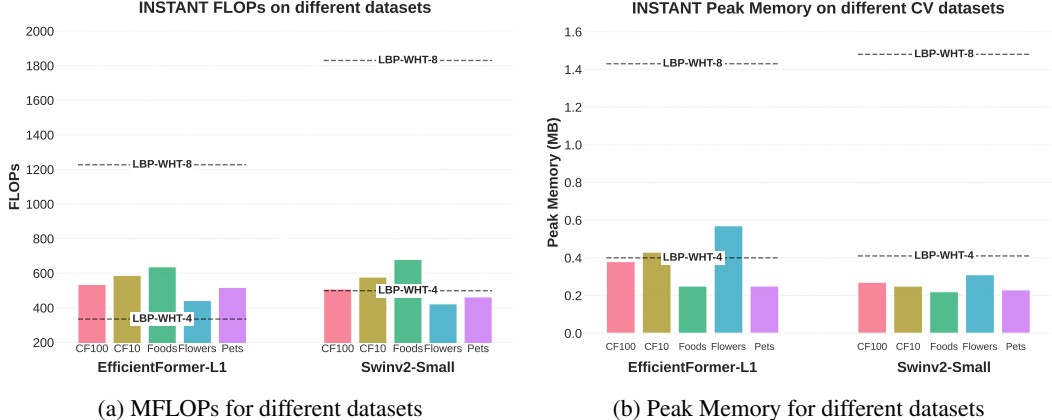

(a) MFLOPs for different datasets      (b) Peak Memory for different datasets

Figure 12: (a) MFLOPs of INSTANT when fine-tuning on 5 datasets with oversampling $p = 5$. There are differences in MFLOPs of each dataset, however, in all datasets, INSTANT has a lower MFLOPs compared to LBP-WHT-8. (b) Peak memory of INSTANT when fine-tuning on 5 datasets with oversampling $p = 5$. The peak memory on different datasets slightly vary, and in all datasets, INSTANT has lower peak activation memory than LBP-WHT-8.

### K.5 MORE SAMPLES ON LOW-RANK CHARACTERISTIC

Fig. 11 describes the low-rank characteristic of various samples through various layers of different architectures. In many layers, a large amount of information can be kept on only a few eigenvalues, proving that the compression tensor of INSTANT can work effectively on many blocks of the model.

### K.6 EFFICIENT TRAINING-AWARE SUBSPACES

Fig. 12 and Fig. 13 illustrate the variation of FLOPs and peak memory between different datasets when we fine-tune with INSTANT. In CV tasks (Fig. 12), INSTANT with $\epsilon = 0.95$ and $p = 5$ has a comparable overhead to LBP-WHT-4, but our performance is better (Tab.1). Compared to LBP-WHT-8, our overhead is extremely smaller in every dataset with a trade-off of negligible accuracy.

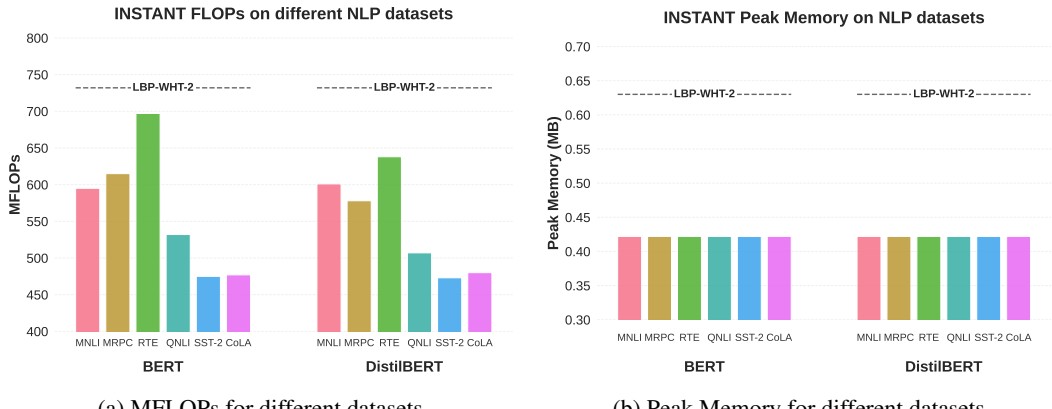

(a) MFLOPs for different datasets   (b) Peak Memory for different datasets

Figure 13: (a) MFLOPs of INSTANT when fine-tuning on 6 datasets with oversampling $p = 7$. There are differences in MFLOPs of each dataset, however, in all datasets, INSTANT has a lower MFLOPs compared to LBP-WHT-2. (b) Peak memory of INSTANT when fine-tuning on 6 datasets with oversampling $p = 15$. The peak memory on different datasets is quite similar, and in all datasets, INSTANT has lower peak activation memory than LBP-WHT-2.

The higher overhead of Flowers is possibly due to its high variance samples, which require larger subspaces $\mathbf{P}$, $\mathbf{Q}$ to capture sufficient meaningful information of tensors. In language tasks (Fig. 12), INSTANT with $\epsilon = 0.95$ and $p = 7$ or $p = 15$ outperform LBP-WHT-2 in every dataset. It is noticeable that activation storage in every dataset is similar. This is possibly due to the strongly low-rank characteristic of activation, especially in language tasks.

## L  LARGE LANGUAGE MODELS (LLMS) USAGE

We utilized an LLM as a support tool in preparing this paper. Its role was limited to:

- polishing the clarity and flow of writing (but we do not use it to generate new text)
- assisting with retrieval and discovery, such as identifying relevant prior work and commonly used methods

All scientific design choices, methodological decisions, implementation, data analysis, and interpretation of results were made solely by the authors.

The LLM did not contribute novel ideas or conduct experiments; it was used only as an assistant for writing and literature awareness.

