# OpenReview forum: "INSTANT: Compressing Gradients and Activations for Resource-Efficient Training"
_ICLR.cc/2026/Conference — ICLR 2026 Poster_

### Official Review · Reviewer_9WbD · 2025-10-29

**Soundness:** 3
**Presentation:** 3
**Contribution:** 2
**Rating:** 4
**Confidence:** 3

**Summary:**

This paper proposes a new distillation technique called grafting, aimed at improving the generalization of autoregressive language models. The grafting strategy integrates sequence trees generated at multiple temperatures into a single distillation target. The paper emphasizes the balance between model compression and generalization, addressing the mode-covering behavior crucial for building more generalizable models. The experimental validation demonstrates the method's effectiveness across several datasets, particularly CIFAR-10 and CIFAR-100.

**Strengths:**

Originality: The grafting strategy is an original approach for balancing model compression and generalization in language model distillation.

Experimental Validation: The method is validated across several datasets, and the results show some potential for improving model performance while reducing computational and memory costs.

Clarity: The paper is well-organized and clearly explains the experimental setup and methodology.

**Weaknesses:**

Lack of Comparison with Related Work: While the paper mentions the error accumulation problem in other activation compression methods, it does not provide a sufficient comparison with existing methods like Sakr & Khailany’s ESPACE or Yang et al.’s LBP-WHT in terms of accuracy and performance. Without this comparison, the claims about grafting’s effectiveness remain unclear.

Limited Experimental Setup: The experiments are mainly focused on simple image classification tasks (e.g., CIFAR-10, CIFAR-100). These datasets may not fully demonstrate the method's potential in more complex or real-world scenarios.

Unaddressed Limitations: The paper does not sufficiently explore some of the limitations of the proposed method, including the computational cost of SVD and its scalability to larger models or long sequences.

**Questions:**

Could you provide a more detailed comparison between grafting and existing methods like Sakr & Khailany’s ESPACE or Yang et al.’s LBP-WHT, particularly in terms of accuracy and computational efficiency?

How do you plan to scale the grafting method for larger models or longer sequences, and what are the potential challenges?

Could you provide further experiments on more complex datasets or real-world tasks to validate the method’s generalizability?

---

> ### Author Response · Authors · 2025-11-20
> **Restatement of existing experiments and extra baselines for comparisons (Response 1/n)**
>
> Thank you for the feedback. We would like to kindly point out that our paper presents a compression method that projects gradients and activations into a low-rank subspace and performs computation within these compressed representations. As such, we neither perform distillation nor grafting. We also do not present a "grafting strategy integrates sequence trees generated at multiple temperatures into a single distillation target" .
>
> ***→ W1 & Q1. Could you provide a more detailed comparison between grafting and existing methods like Sakr & Khailany’s ESPACE or Yang et al.’s LBP-WHT, particularly in terms of accuracy and computational efficiency?***
>
> Please note that we are not performing ‘grafting’. In terms of comparison to related methods, such as LBP-WHT, we kindly refer the reviewer to Section 4 *(line 270, main paper)*. Since LBP-WHT is our main competitor, we provided a detailed comparison between our method and LBP-WHT in terms of both accuracy and computational efficiency *(Tab.1, line 340 and Tab.2 line 378)*.
>
> Regarding ESPACE, we would also like to clarify why ESPACE was not included in our original set of baselines. ESPACE primarily focuses on compressing activations only *(line 96-100 main paper)*, whereas our main objective is to jointly compress both activations and activation gradients. For this reason, ESPACE targets a somewhat different problem setting from ours, which is why it was not selected as a main comparison in the initial submission. However, we provide an extra comparison with ESPACE below. We also provide extra baseline: *Random -* which is INSTANT using random subspaces $P, Q$  for compressing activation $x$ and activation gradient $g_y$ to highlight the importance and efficiency of our calibration.
>
> | **Partial Finetuning** | **CIFAR10** | **CIFAR100** | **Forward MFLOPs** | **Backward MFLOPs** | **Activation Memory (MB)** |
> | --- | --- | --- | --- | --- | --- |
> | ESPACE – N/2 | **95.23** | 78.41 | 414 | 827 | 0.96 |
> | ESPACE – N/4 | 94.21 | 76.26 | 212 | 414 | 0.48 |
> | Random – N/2 | 86.52 | 73.13 | 742 | 827 | 0.96 |
> | Random –N/4 | 64.97 | 46.02 | 742 | 414 | 0.48 |
> | INSTANT-0 | *94.66* | 77.64 | 742 | 270 | 0.16 |
> | INSTANT-5 | 95.07 | *78.65* | 742 | 475 | 0.38 |
> | INSTANT-7 | **95.23** | **79.01** | 742 | 544 | 0.45 |
> | Vanilla | 95.23 | 79.28 | 742 | 1484 | 1.95 |
>
> | **Full Finetuning** | **CIFAR10** | **CIFAR100** | **Forward MFLOPs** | **Backward MFLOPs** | **Activation Memory (MB)** |
> | --- | --- | --- | --- | --- | --- |
> | ESPACE – N/2 | 93.16 | 74.56 | 2347 | 4694 | 4.29 |
> | ESPACE – N/4 | 51.18 | 35.62 | 1173 | 2347 | 2.14 |
> | Random – N/2 | 13.47 | 3.66 | 2264 | 4694 | 4.29 |
> | Random – N/4 | 10.00 | 9.26 | 2264 | 2347 | 2.14 |
> | INSTANT-5 | 96.29 | 82.41 | 2264 | 2107 | 1.98 |
> | INSTANT-10 | *96.48* | *83.05* | 2264 | 2491 | 2.73 |
> | INSTANT-15 | **96.85** | **83.56** | 2264 | 2884 | 3.45 |
> | Vanilla | 96.99 | 84.84 | 2264 | 4528 | 18.46 |
>
> The experimental results demonstrate that INSTANT significantly outperforms both ESPACE and the *Random* projection method, where the compression rank is set to $\frac{N}{2}$ or $\frac{N}{4}$ , with $N$ representing the input dimension. This highlights that INSTANT effectively requires a compact yet precise low-rank space to retain gradient information for accurate backpropagation through the network. Notably, in the case of full fine-tuning, the *Random* projection method fails to converge, as the gradients struggle to propagate through multiple layers, resulting in significant error accumulation due to the lack of a well-defined low-rank space. These extra experimental results are updated to the main paper *(line 1338, line 1377).*
>
> Moreover, understanding the concerns about the comparison between our method and other optimizer gradient compression methods (GaLore, CompAct), we also add extra baselines in the Appendix H *(line 1129)* and analyze the strengths, weaknesses of these methods, and suggest the combination of INSTANT with these methods.

---

> > ### Author Response · Authors · 2025-11-20
> > **(Continue) Restatement of existing experiments and extra baselines for comparisons (Response 2/n)**
> >
> > ***→ W2. Limited Experimental Setup: The experiments are mainly focused on simple image classification tasks (e.g., CIFAR-10, CIFAR-100). These datasets may not fully demonstrate the method's potential in more complex or real-world scenarios.***
> >
> > In our experimental setup, we do not restrict ourselves to simple image classification tasks. We also conduct extensive experiments on BERT-like models using the GLUE benchmark to evaluate performance on natural language processing tasks *(Tab.2 line 378)*. These benchmarks, both for vision and natural language processing, are widely adopted in the community for assessing compression methods, and many prior works—such as LBP-WHT, PocketEngine, On-Device Training, LoRA, and GaLore—use the same evaluation protocols.
> >
> > ***→ W4 & Q2. How do you plan to scale the grafting method for larger models or longer sequences, and what are the potential challenges?***
> >
> > In Appendix L *(line 1373)*, we conduct additional experiments to evaluate the behavior of INSTANT on larger models such as ViT-B/32 *(line 1385)* and TinyLlama *(line 1433)*. The results indicate that, on ViT-B/32, INSTANT achieves performance comparable to standard training while reducing both activation memory and computational cost by a factor of 5. For TinyLlama, the performance gap is larger; however, our method still yields substantial savings in activation memory and computational complexity.
> >
> > While the results are encouraging, we identify some potential challenges that we believe should be addressed in future work to further scale INSTANT:
> >
> > - **Error accumulation in deeper models.** As model depth increases, error stacking may become more pronounced. This could necessitate more frequent calibration updates or the use of a larger energy threshold and oversampling factor in order to reliably capture an appropriate low-rank subspace.
> > - **Limited calibration for very large models.** As shown in Appendix L.4 *(line 1434)*, when scaling to models with around one billion parameters, the performance gap between INSTANT and vanilla training widens. An explanation is that large models have higher capacity, so a small number of calibration batches may not provide sufficiently rich statistics to construct an accurate low-rank projection, ultimately making it harder to capture the optimal low-rank space.
> >
> > ***→ Q3. Could you provide further experiments on more complex datasets or real-world tasks to validate the method’s generalizability?***
> >
> > Thank the reviewer for this suggestion. We finetuned EfficientFormer-L1 on Places-365 datasets, which contains 1.8M images and is more challenging than ImageNet (models often perform worse on Places-365 than ImageNet).
> >
> > |  | **Accuracy** | **Backwrard MFLOPs** | **Activation Memory** |
> > | --- | --- | --- | --- |
> > | Vanilla | 55.30 | 1484 | 1.95 |
> > | Gradient Filtering | 9.2 | 24 | 0.04 |
> > | LBP-WHT-2 | 50.92 | 95 | 0.12 |
> > | LBP-WHT-4 | 53.27 | 335 | 0.40 |
> > | LBP-WHT-8 | 54.67 | 1227 | 1.43 |
> > | INSTANT-0 | 54.32 | 388 | 0.37 |
> > | **INSTANT-5** | **54.57** | **567** | **0.60** |
> > | INSTANT-7 | 54.55 | 606 | 0.66 |
> >
> > The table above demonstrates that INSTANT remains highly effective even with large datasets such as Places-365. All three versions of INSTANT show only a minor reduction in performance compared to vanilla training, despite achieving a threefold reduction in both computational cost and memory usage. When compared to LBP-WHT, INSTANT offers slightly better performance while maintaining the same memory and computational budget. In contrast, Gradient Filtering fails to converge, yielding poor results. Thank the Reviewer for this suggestion. We added these experiments to our main paper *(Appendix L.3, line 1397).*

---

> > > ### Author Response · Authors · 2025-11-20
> > > **Analysis of Calibration overhead and Restatement of existing experiments (Response 3/n)**
> > >
> > > ***→ W3.1. Unaddressed Limitations: The paper does not sufficiently explore some of the limitations of the proposed method, including the computational cost of SVD.***
> > >
> > > We propose a calibration-based algorithm for constructing the projection, specifically designed to mitigate the computational cost of SVD. Importantly, under our calibration scheme, SVD is performed only during the calibration phase, and within each calibration, it is applied at most once per layer, thereby only introducing a small computational cost. The details of this procedure are provided in Appendix B.3 *(line 762)*, in which we proved the calibration cost to be 1/40 computational cost of backpropagation.
> > >
> > > Considering our targets of reducing both peak memory and FLOPs, our calibration needs to be efficient:
> > >
> > > - **Calibration does not increase peak memory** thanks to on-policy processing several batches at once, and a smaller batch setup (Appendix B.3, *line 766*).
> > > - **Small calibration computation** thanks to long-term subspaces utilization, which is proved via the following analysis of the ratio between calibration FLOP $(flop_c)$ and training FLOP $(flop_t)$.
> > >
> > > **Proof of small computation overhead of calibration:**
> > >
> > > - **Experiment Setup:** 5 calibration iterations for each $N_t = 200$ training iteration (as in our experiments). For the sake of simplicity, our linear layer: $y = x\cdot w^T$ with $x \in \mathbb{R}^{B \times L \times C_x}, w \in \mathbb{R}^{C_y \times C_x}, y \in \mathbb{R}^{B \times L \times C_y}$.
> > > - FLOP training: $flop_t = 200\cdot (6\cdot B\cdot L\cdot c_x\cdot C_y)$  (forward and backward pass)
> > > - FLOP calibration (Algorithm 1, *line 776 in main paper*): $f_c=5 \cdot (6\cdot B\cdot L\cdot C_x\cdot C_y)$  (step 6) $+5\cdot (2\cdot B\cdot L^2\cdot (C_x + C_y))$ (step 7) $+ [4/3 \cdot L^3 + 4/3 \cdot L^3]$ (SVD cost, step 13)
> > > - Ratio: $f_c / f_t= \frac{30\cdot B\cdot L\cdot C_x\cdot C_y+ 10\cdot B\cdot L^2\cdot (C_x+C_y)+8/3\cdot L^3)}{1200\cdot B\cdot L\cdot C_x\cdot C_y}$ ~ $\frac{1}{40} (L < min(C_x, C_y))$
> > >
> > > Therefore, FLOP of calibration is small compared to FLOP of training (Eg, in table 1, *line 340* of the main paper, INSTANT-5 with reported **475** MFLOPs for training on EfficientFormerL1, requires an extra **37** MFLOPs for calibration)
> > >
> > > Therefore, only reporting training FLOPs still reflects a fair comparison with other methods. However, this calibration FLOPs is small but not negligible. Thank the Reviewer for this suggestion. We added this analysis to the main paper to make our method clearer *(line 810)*.

---

> > > > ### Comment · Reviewer_9WbD · 2025-11-25
> > > > **Response to the rebuttal**
> > > >
> > > > The authors have addressed most of my concerns, so I have raised my score.

---

### Official Review · Reviewer_jz4Y · 2025-10-30

**Soundness:** 2
**Presentation:** 3
**Contribution:** 3
**Rating:** 6
**Confidence:** 4

**Summary:**

This paper proposes a compression method called INSTANT. INSTANT is a method that projects the activations and gradients to lower ranks, based of analysis that is done at regular intervals during training. The authors show the benefits of compressing both the gradients and activations during training by showing reduction in FLOPs and training time (in some cases).

**Strengths:**

1. The paper is  tackling both memory and computational bottlenecks.  Activation compression for memory saving has been explored previously, but the idea of also compressing the activation gradient ($g_y$) to reduce backpropagation FLOPs is a useful contribution.

2. The authors provide empirical evidence (Fig. 3) to support their intuition that activation gradients are inherently low-rank, justifying their compression approach.

3. NSTANT's  SVD-based approach is data-driven. This allows it to generalize more effectively to other modalities, as demonstrated by its strong performance on NLP tasks (Table 2) where previous works (eg LBP-WHT) struggles.

4. The authors validates the method's usability by reporting significant wall-clock speedups (2x to 12.5x) on resource-constrained edge CPUs.

**Weaknesses:**

I believe, currently this paper's primary weaknesses lie in the evaluation of its computational claims and the transparency of its overhead costs.

**Omission of Wall-Clock Training Time on GPUs:**

1. The paper's main results (Tables 1 & 2) were generated on an NVIDIA V100 GPU but only report FLOPs, not wall-clock training time.

a) The authors' justification (Section 4.1) that "FLOPs... [are] unaffected by implementation details" is a weak defense. A reduction in FLOPs does not guarantee a proportional reduction in training time, especially on GPUs. Modern deep learning libraries (like cuDNN) are highly optimized for large, dense matrix multiplications, and the low-rank operations introduced by INSTANT may not be as implementation-efficient, thus creating a bottleneck.

b) The authors themselves admit a discrepancy in Appendix I, stating, "The (12x) time reduction is not comparable to (17x) FLOP reduction," even on a CPU. This gap is likely to be even larger on a GPU, and the lack of this data makes it difficult to assess the practical speedup in a typical training environment.

**Cost and Nature of the "Static" Subspace:**

2. The term "static" subspace (used in the contribution list and abstract) could be misleading. The subspace is not fixed; it is "periodically" recalibrated every $N_t$ steps (e.g., $N_t=50$ or $N_t=200$).

a) The computational cost of this recalibration (Algorithm 1) appears to be non-trivial and is not accounted for in the reported per-step FLOP savings. This calibration requires running multiple batches and performing SVD for every layer being compressed. This overhead could significantly diminish the overall wall-clock time savings.

b) The method's stability relies on an "oversampling" hyperparameter $p$, which is introduced to "reduce information loss when the core bases change" between calibrations (Section 3.2, Fig. 6). This suggests the "static" assumption is fragile and introduces another sensitive hyperparameter that must be tuned, adding to the method's complexity.

**Questions:**

1. Could the authors please provide the total wall-clock training time (not just backward time) for the main V100 GPU experiments in Tables 1 and 2? This is essential for evaluating the practical speedup of INSTANT in a standard training scenario.

2. How is the computational cost of the periodic calibration step (Algorithm 1) factored into the total FLOPs reported? Could you provide an analysis of this overhead, for instance, as a percentage of total training time or FLOPs?

3. Given that the subspace is recalibrated every $N_t$ steps, would "periodically updated" or "cached" be a more accurate description than "static"?

4. The oversampling parameter $p$ appears critical for performance (Fig. 6, Fig. 8). How sensitive is the method to the choice of $p$ and the calibration frequency $N_t$? Does this introduce a significant hyperparameter tuning burden?

5. Following up on the 17x FLOP vs. 12x time gap noted in Appendix I: What is the authors' hypothesis for the performance gap on a V100, where highly optimized dense matrix multiplication kernels are the standard?

---

> ### Author Response · Authors · 2025-11-20
> **Analysis of Training time on GPU and hypothesis about the time increase (Response 1/n)**
>
> We appreciate the Reviewer’s comprehensive comments. As you suggested (*W1 & Q1 & Q5*), we did experiments to get the training time of INSTANT on V100 GPU.
>
> ***→ W1: “Omission of Wall-Clock Training Time on GPUs”***
>
> ***→ Q1: “Could the authors please provide the total wall-clock training time (not just backward time) for the main V100 GPU experiments in Tables 1 and 2?”***
>
> We conducted experiments on the framework of MMCV, model EfficientFormerL1, with datasets CIFAR10, and with all methods as provided in Tab.1 *(line 340, main paper)* (LBP-WHT, Gradient Filtering, INSTANT (ours), Vanilla). We observed similar results of CIFAR100. Due to the limited time of Rebuttal, we just ran experiments with these 2 datasets.
>
> **Last Layer+ Full Finetuning EfficientFormerL1 on CIFAR10– Time reported for training 1 epoch with V100.**
>
> |  |  | **Last** | |  | **Full** | |
> | --- | --- | --- | --- | --- | --- | --- |
> | Method/Time(s)  | Calibration* | Backward** | Total epoch*** | Calibration* | Backward** | Total epoch*** |
> | Vanilla | *0* | *6.11* | *73.6* | *0* | *12.66* | *110.8* |
> | Gradient Filtering | 0 | 7.77 | 74.0 | 0 | 15.07 | 94.2 |
> | LBP-WHT-2 | 0 | 7.99 | 79.8 | 0 | 19.6 | 146.4 |
> | LBP-WHT-4 | 0 | 9.02 | 123.8 | 0 | 18.55 | 246.0 |
> | LBP-WHT-8 | 0 | 10.45 | 291.4 | 0 | 20.5 | 625.0 |
> | INSTANT-0 | 2.16 | **8.8** | **74.2** | 4.99 | **17.55** | **116.6** |
> | INSTANT-5 | 2.2 | 9.01 | 74.0 | 4.98 | 17.68 | 120.0 |
> | INSTANT-7 | 2.22 | 8.73 | 74.2 | 4.98 | 17.17 | 120.2 |
>
>  *( * ) **Calibration time** is **total additional time** used for creating periodically updated subspaces during 1 training epoch ( ** ) **Backward time** is total time of `loss.backward()` during 1 training epoch. ( *** ) **Epoch time** is the **total running time** of training 1 epoch, including calibration time (with INSTANT), forward pass, activation savings, loss calculation, backward pass, optimizer update,….. These reported numbers are averaged over 10 training epochs.*
>
> On GPU V100, **INSTANT’s backward time is 1.4x compared to Vanilla backwards time**. The total training time (epoch time) of INSTANT is slightly higher than Vanilla. In short, INSTANT’s FLOP reductions are not converted into time reduction like training on CPU.
>
> ***→ Q5: “Following up on the 17x FLOP vs. 12x time gap noted in Appendix I: What is the authors' hypothesis for the performance gap on a V100”***
>
> As shown in the table, contrary to time reduction on CPU *(line 427, main paper)*, INSTANT slightly increases training time on GPU V100. We believe this gap is due to that CUDA is optimized for large matrices (as the Reviewer suggested), and kernel launching overhead.
>
> - **CUDA optimized for large matrices** is the main reason for the slower INSTANT implementation on the GPU. A GPU with multiple cores is designed for parallel computations of large operations. Splitting a large operation into three smaller ones lowers the utilization of GPU cores and leads to overhead.
> - **Kernel launching overhead and Memory bandwidth**: INSTANT low-rank $y = (x \cdot P^T)\cdot(P\cdot w^T)$ launches 3 CUDA kernels for 3 multiplications, while Vanilla $y = x \cdot w^T$ requires launching only 1 CUDA kernel.  Moreover, specialized memory types like GDDR6 or HBM (High Bandwidth Memory) lead to rapid data transfers and access within the GPU, which reduces the time for large matrix loading.
>
> **Why INSTANT gains on CPU?** Different from parallel computing on GPU with multiple cores, CPU has much smaller number of cores so it cannot compute a large computation at once. Moreover, in Von Neumann architecture, CPU needs to access data from cache/memory. Small matrices can be loaded from cache, which reduces time, while loading a large matrix from memory takes additional time.
>
> In short, both devices have the gain of smaller arithmetic multiplication, but this gain is small compared to the extra cost of kernel launching overhead and memory bandwidth.
>
> We would like to thank you for the insightful question, which helps highlight the applicability of our work from additional perspective. We updated our paper *(line 431, line 1263)* to include this discussion. Nevertheless, we we will definitely focus on this topic in future work.

---

> ### Author Response · Authors · 2025-11-20
> **Calibration overhead and choices of hyperparameters (Response 2/n)**
>
> ***→  "W2 & Q2: "How is the computational cost of the periodic calibration step (Algorithm 1) factored into the total FLOPs reported? Could you provide an analysis of this overhead, for instance, as a percentage of total training time or FLOPs?""***
>
> Considering our targets of reducing both peak memory and FLOPs, our calibration needs to be efficient:
>
> - **Calibration does not increase peak memory** thanks to on-policy processing several batches at once, and smaller batch setup *(line 766 main paper)*
> - **Small calibration computation** thanks to long-term subspaces utilization, which is proved via the following analysis of the ratio between calibration FLOP $(flop_c)$ and training FLOP $(flop_t)$ *(line 810 main paper).*
>
> **Proof of small computation overhead of calibration:**
>
> - 5 calibration iterations for each $N_t = 200$ training iteration (as in our experiments). For the sake of simplicity, our linear layer: $y = x\cdot w^T$ with $x \in \mathbb{R}^{B \times L \times C_x}, w \in \mathbb{R}^{C_y \times C_x}, y \in \mathbb{R}^{B \times L \times C_y}$.
> - FLOP training: $flop_t = 200\cdot (6\cdot B\cdot L\cdot c_x\cdot C_y)$  (forward and backward pass)
> - (Algorithm 1 in main paper): FLOP_calibration: $f_c=5 \cdot (6\cdot B\cdot L\cdot C_x\cdot C_y)$  (step 6) $+5\cdot (2\cdot B\cdot L^2\cdot (C_x + C_y))$ (step 7) $+ [4/3 \cdot L^3 + 4/3 \cdot L^3]$ (SVD cost, step 13)
> - Ratio: $f_c / f_t= \frac{30\cdot B\cdot L\cdot C_x\cdot C_y+ 10\cdot B\cdot L^2\cdot (C_x+C_y)+8/3\cdot L^3)}{1200\cdot B\cdot L\cdot C_x\cdot C_y}$ ~ $\frac{1}{40} (L < min(C_x, C_y))$
>
> Therefore, FLOP of calibration is small to FLOP of training (Eg. in Tab.1, INSTANT-5 with reported 475M training FLOPs on EfficientFormerL1 *(line 351 main paper)* requires extra 37 MFLOPs for calibration)
>
> Therefore, only reporting training flop still reflects a fair comparison with other methods. However, this calibration FLOPs is small but not negligible. Thank the Reviewer for this suggestion. We added this analysis to the main paper to make our method clearer *(line 419, line 810 of main paper)*.
>
> ***→ “Q3: Given that the subspace is recalibrated every  steps, would "periodically updated" or "cached" be a more accurate description than "static"?”***
>
> Yes, “periodically updated” is more accurate. Our purpose of writing “static” is to indicate the slow change of the compression tensor over time (static over $N_t =200$ training iterations). However, “static” is not exactly accurate. We appropriately thank the reviewer for their suggestion. We changed this in the main text.
>
> ***→ “Q4: The oversampling parameter  appears critical for performance (Fig. 6, Fig. 8). How sensitive is the method to the choice of p and the calibration frequency? Does this introduce a significant hyperparameter tuning burden?”***
>
> - **Regarding to oversampling hyperparameter $p$**, with small $p (0,1,2,3)$, the subspace selection varies significantly so the performance changes much. For example, due to low-rank activation and gradient, subspaces $P$ (or $Q$) with 1 vector can achieve good approximation (reach $\epsilon=0.95$) (as shown in Figure 11 in main paper). Oversampling with $p = 5$ will result to subspaces with 6 vectors, which significantly affect the performance. When $p$ becomes larger, they do not capture the core subspace more so performance does not change much (subspace with 6 vectors $(p=5)$ is comparable to subspace with 11 vectors $(p=10)$). We conducted experiments to see the sensitivity of $p$. As shown in our Figure.6, *(line 445)* and Figure.8b *(line 1095)*, the performance tends to become saturated when increasing $p$. Therefore, with small $p$, INSTANT is sensitive to that value, but if the resource we have is bigger, $p$ is not so sensitive.
> - **Regarding to calibration frequency**, smaller $N_t$ (update subspace more frequently) is better as shown in the Appendix. However, smaller $N_t$ results in more calibration overhead, and its computational cost may not be negligible anymore. To find an appropriate calibration frequency, we do empirical study as shown in Tab.1 *(line 340)* , Tab.2 *(line 378)* and Figure 8 *(line 1095)*.
>
> In short, these choices of $p$ and $N_t$ depend on resource we want. Like other hyperparameters, we need to tune to find the appropriate numbers. However, these choices are stable within task, e.g the choices of $p=5$, $N_t=200$ perform well in 5 image datasets.

---

### Official Review · Reviewer_FBNs · 2025-10-31

**Soundness:** 3
**Presentation:** 4
**Contribution:** 1
**Rating:** 2
**Confidence:** 5

**Summary:**

This paper proposes INSTANT, a training-time compression method that projects activations and gradients into low-rank subspaces and performs backpropagation in those compressed representations. The authors describe (i) an SVD-based calibration to build per-layer projectors, (ii) projection/truncation with an energy threshold + oversampling, and (iii) a low-rank backward algorithm. Experiments show large reductions in activation memory and FLOPs across Transformer and CNN models with small accuracy drops.

**Strengths:**

* Solid empirical evaluation across modalities (vision & NLP) and architectures (Transformers, CNNs).
* Clear exposition, reproducibility-minded appendices and pseudocode.
* Practical relevance: reduces activation memory and backward FLOPs as demonstrated in their experiments, with sensible ablations (oversampling, calibration frequency, rank choices).
* Sound theoretical guarantee for stable low-rank training. Their analysis shows that SVD-based projections minimize reconstruction error and that gradient approximation error remains bounded through depth, vanishing as the retained energy ε → 1.
* Includes deployment-relevant experiments (edge device / Raspberry Pi).

**Weaknesses:**

1. **Severe overlap with prior work (CompAct, NAACL 2025, publicly available on arXiv since Oct 2024):**
Conceptually highly similar and structurally parallel to CompAct [1], differing mainly in projection construction and calibration choices, without mentioning CompAct at all, despite the latter being published and publicly available months before ICLR submission.
Both papers:
* Compress activations via low-rank projections during the forward pass.
* Compute gradients in the compressed subspace and decompress for weight updates.
* Aim to reduce memory and optimizer overhead jointly.
* Demonstrate scaling benefits on LLaMA and BERT-like models.
The algorithmic structure of INSTANT (forward compression → compressed backward → decompression for update) matches CompAct’s Algorithms 1–3. The overlap extends to terminology (“projected activations,” “reduced optimizer states”), theoretical justification, and empirical evaluation.

2. **Novelty and contribution are overstated:**
While INSTANT adds implementation refinements, such as calibration-based rank selection and changes the choice of projection matrix, these are engineering extensions rather than conceptual advances. The claim of being the “first to jointly compress activations and gradients” is false, due to the overlap with prior work.

3. **Lack of comparison:**
The experiments do not compare with the most appropriate relevant works like GaLore [2] or any of the myriad of works that followed it (VeLORA [3] ,Grass [4] ,WeLore [5]...). Reported baselines are insufficient to establish novelty or superiority.


References
* [1] CompAct: Compressed Activations for Memory‑Efficient LLM Training – Shamshoum et al., NAACL 2025, arXiv:2410.15352v1.
* [2] GaLore: Memory‑Efficient LLM Training by Gradient Low‑Rank Projection – Zhao et al., ICML 2024, arXiv:2403.03507.
* [3] VeLORA: Memory Efficient Training using Rank‑1 Sub‑space Activations – Miles et al., NeurIPS 2024 Poster, algorithm for compressing activations into 1-D subspace.
* [4] Grass: Compute Efficient Low‑Memory LLM Training with Structured Sparse Gradients – Muhamed et al., arXiv:2406.17660.
* [5] WeLore: Weight Low‑Rank Projection for Memory‑Efficient Fine‑Tuning – Jaiswal et al., ICLR 2025, arXiv:2407.11239.

**Questions:**

* Were the authors aware of CompAct (NAACL 2025) at submission time?
* What specific conceptual or methodological innovations distinguish INSTANT from CompAct beyond the choice of projection matrix?
* Can you provide a direct empirical comparison to CompAct under matched conditions (e.g., same model, rank, dataset)?
* How do the claimed FLOP reductions compare to a random projection baseline?

**Details Of Ethics Concerns:**

The paper omits citation to a highly similar prior work (CompAct), raising ethical concerns about the faithfulness of its novelty claims and proper credit assignment. Merely googling “LLM activation compression” shows CompAct in the top results, so it is unlikely the paper was missed unintentionally. Despite strong experiments and presentation, if the authors don’t provide a thorough and experimental comparison with this highly overlapping work, I will keep recommending a strong reject.

---

> ### Author Response · Authors · 2025-11-20
> **Addressing Ethics Concerns: INSTANT is different from CompAct (Response 1/n)**
>
> First of all, we appreciate your detailed comments. We would like to clarify your concerns raised in ***W1, Q2*** and the ***ethical concerns***: Our work is **completely original and does not include any overlaps, plagiarized parts, or any kind of misconduct**. We understand the reviewer’s concerns, but we do not take such accusations lightly, as we believe scientific integrity to be of utmost importance.
>
> ***→ W1 & Q2 & “Ethics Concerns”: Severe overlap with prior work (CompAct, NAACL 2025, publicly available on arXiv since Oct 2024):***
>
> Our work and the one referenced in the review [1] (CompAct, NAACL 2025) are related, as both follow similar research directions of compression approaches that address computational and memory bottlenecks during training. In this space, there are several classes of methods that leverage gradient compression, activation compression, and/or compression in the optimizer state. In this context, our work compared to CompAct (and other related works) differs in several aspects. Specifically, INSTANT focuses on compressing the low-rank gradient of the activation gradient during backpropagation. Mathematically, the difference between CompAct and INSTANT in the backpropagation is:
>
> - CompAct:                     $\mathbf{\hat{g}_w}= \mathbf{g_y}^\top \cdot \mathbf{\hat{x}}$               and             $\mathbf{g_x} = \mathbf{g_y} \cdot \mathbf{w}$
> - INSTANT:         $\mathbf{g_w} = \left(\mathbf{g_y}^\top \cdot \mathbf{P}^ \top \right) \cdot \left( \mathbf{P} \cdot \mathbf{x} \right)$ and    $\mathbf{\tilde{g}_x} = \mathbf{Q}^\top \cdot \left( \left( \mathbf{Q} \cdot \mathbf{g_y} \right) \cdot \mathbf{w} \right)$
>
> We compress both the equation for calculating weight activation and gradient activation to much lower the computational cost during backpropagation. This is the novelty of our work, separating our work from GaLore, CompAct, and Grass, which focus on the low-rank of the weight gradient $\left(\mathbf{g_w}=\frac{dL}{dw}\right)$ in the optimizer. In fact, our work **does not integrate** **the optimizer state.** As such, our work differs from all works that perform compression of the optimizer state, including CompAct. The difference between INSTANT and CompAct is:
>
> | Aspect | CompAct | INSTANT (ours) |
> | --- | --- | --- |
> | Compression target | Activation + Weight Gradients + Optimizer States | **Activation + Activation Gradients only** |
> | Backpropagation | Only weight gradients computation are in low-rank space | **Both weight and activation gradients are computed in a compressed space** |
> | Source of savings | Optimizer state and activation memory | **Backward FLOP reduction + activation memory** |
> | Projection | Heuristic | **Calibrated low-rank subspace based on sufficient statistics with proofs** |

---

> ### Author Response · Authors · 2025-11-20
> **(Continue) INSTANT is different from CompAct(Response 2/n)**
>
> In detail, the differences further extend to:
>
> - **“Compute gradients in the compressed subspace and decompress for weight updates”:**
>
>     There are two main matrix multiplications during the backpropagation process: the weight gradient $g_w$ for updating weights and the activation gradient $g_x$ for propagating back to the previous layer. CompAct only computes the $g_w$ in the low-rank space, while keeping $g_x$ in the original dimensionality (as shown in Fig.2 and Alg.2 of [1]). In contrast, *INSTANT does both multiplications in the low-rank space to enormously reduce the computational cost in the backpropagation process, as shown in our equations (4) and (5).*
>
> - **“Aim to reduce memory and optimizer overhead jointly”:**
>
>     In this work, we focus only on reducing memory usage and computational cost during backpropagation, **without modifying the optimizer** state like CompAct (and Galore[2]). Although our low-rank projection mechanism could in principle be applied to the optimizer as well, this direction is beyond the scope of the present study.
>
>     *Our work leverages the low-rank characteristics of the activation gradient.* This is a novel approach, which is not considered in other work, and as such is different from other related works (ComPact, Galore, WeLore, VeLore, Grass).
>
> - **“Demonstrate scaling benefits on LLaMA and BERT-like models. The algorithmic structure of INSTANT (forward compression → compressed backward → decompression for update) matches CompAct’s Algorithms 1–3. T... and theoretical justification,...”:**
>
>     As we noted, our work differs from CompAct in that we do not target compression of the optimizer state, which is the central focus of CompAct. As such, the algorithms also differ. This also extends to our theoretical justification.
>
> - “**The overlap extends to terminology (“projected activations,” “reduced optimizer states”)”**
>
>     We don’t use this specific construct/terminology. As our method focuses on activations that have been low-rank projected, we use the words to explain it. We do not use “reduced optimizer states” as our method doesn’t apply to that step. We only mention this when discussing the mechanics of a related work, GaLore, which uses it.
>
> - **“...and empirical evaluation..”:**
>
>     We follow a standard empirical evaluation setup from previous work (Galore) that includes  LLaMA and BERT-like models and datasets, which also seems to be the case for CompAct. More importantly (and differently), however, we extend our setup and report experiments on vision tasks to further assess the effectiveness and generality of our method (similar to LBP-WHT).
>
> - **“Compress activations via low-rank projections during the forward pass:”**
>
>     While both INSTANT and CompAct compress the activation for lower memory storage, INSTANT differs from CompAct in how we construct a low-rank projection. This is, in fact, one of our main contributions, as we propose a novel method for calibrating the proper low-rank projection tensor.
>
>
> ***→ Q1. Were the authors aware of CompAct (NAACL 2025) at submission time?***
>
> We are aware of it, also at submission time. However, we considered it as a variant of GaLore in the direction of gradient compression, which has been discussed in our related work. Our work is independent from CompAct, and both have been concurrently developed. Therefore, it is indeed an oversight on our end to not include it in the related work discussion and provide a more detailed comparison. This is the **only** issue on this matter.
>
> To remedy this omission revised and expanded this part, adding further references and explicitly clarifying the distinctions between our approach and the cited papers (*line 10*1 to *line 110, line 267* to *line 269*). We also extended our baseline in the Appendix. H, as shown in the Response 4/n.

---

> > ### Author Response · Authors · 2025-11-20
> > **Address novelty concern and Proof for importance of our mathematical-based projection (Response 3/n)**
> >
> > ***→ W2.1:“The claim of being the “first to jointly compress activations and gradients” is false”***
> >
> > We would like to emphasize that we state **we are the first ones that exploit the low-rank characteristic of the activation gradient in many data distributions**, which is not explored in previous work in compressing weight (WeLore [5]), activation (CompAct [1], VeLORA [3]), and optimizer state & weight gradient (CompAct [1], GaLore[2], Grass[4]).
> >
> > ***→ W2.2: “While INSTANT adds implementation refinements, such as calibration-based rank selection and changes the choice of projection matrix, these are engineering extensions rather than conceptual advances”.***
> >
> > Our two baselines, Gradient Filtering and LBP-WHT, despite compressing both activation and the activation gradient, their methods are based on the low-frequency assumption of input data, and only work well on image tasks.
> >
> > Our calibration-based rank selection strategy is novel in that it **identifies an appropriate low-rank subspaces for both activations and activation gradients, which is built on a mathematical foundation via sufficient statistics (Section 3.2) and clear proofs (Appendix B, C).** This enables INSTANT to project tensors into a more compact space while preserving the essential information, thereby ensuring both efficiency and optimal low-rank representations.
> >
> > ***→ Q4: How do the claimed FLOP reductions compare to a random projection baseline?***
> >
> > **Theoretically, random projection cannot ensure good approximation**, which motivates us to build our adaptive subspaces (Appendix  B.1)
> >
> > **Empirically, random projection leads to performance degradation.** We conducted additional experiments to assess the effectiveness of our calibration-based rank selection. In particular, we compared our method against a random projection baseline and observed that calibration consistently yields better low-rank representations.
> >
> > | Partial Training | CIFAR10 | CIFAR100 | Backward MFLOPs | Activation Memory (MB) |
> > | --- | --- | --- | --- | --- |
> > | Random – N/2 | 86.52 | 73.13 | 827 | 0.96 |
> > | Random – N/4 | 64.97 | 46.02 | 414 | 0.48 |
> > | INSTANT-0 | *94.66* | 77.64 | 270 | 0.16 |
> > | INSTANT-5 | 95.07 | *78.65* | 475 | 0.38 |
> > | INSTANT-7 | **95.23** | **79.01** | 544 | 0.45 |
> > | Vanilla | 95.23 | 79.28 | 1484 | 1.95 |
> >
> > | Full Training | CIFAR10 | CIFAR100 | Backward MFLOPs | Activation Memory (MB) |
> > | --- | --- | --- | --- | --- |
> > | Random – N/2 | 13.47 | 3.66 | 4694 | 4.29 |
> > | Random – N/4 | 10.00 | 9.26 | 2347 | 2.14 |
> > | INSTANT-5 | 96.29 | 82.41 | 2107 | 1.98 |
> > | INSTANT-10 | *96.48* | *83.05* | 2491 | 2.73 |
> > | INSTANT-15 | **96.85** | **83.56** | 2884 | 3.45 |
> > | Vanilla | 96.99 | 84.84 | 4528 | 18.46 |
> >
> > The experimental results demonstrate that INSTANT significantly outperforms the random projection method, where the compression rank is set to $\frac{N}{2}$ or $\frac{N}{4}$ , with $N$ representing the input dimension. This highlights that INSTANT effectively requires a compact yet precise low-rank space to retain gradient information for accurate backpropagation through the network. Notably, in the case of full fine-tuning, the random projection method fails to converge, as the gradients struggle to propagate through multiple layers, resulting in significant error accumulation due to the lack of a well-defined low-rank space.

---

> > > ### Author Response · Authors · 2025-11-20
> > > **Showcase INSTANT efficiency by extra comparisons with CompAct, GaLore (Response 4/n)**
> > >
> > > ***→ W3: “Lack of comparison”: The experiments do not compare with the most appropriate relevant works like GaLore [2] or any of the myriad of works that followed it (VeLORA [3] ,Grass [4] ,WeLore [5]...).***
> > >
> > > In our submission, to evaluate the effectiveness of our method, we chose Gradient-Filter and LBP-WHT as baselines, since these papers also discovered the low-rank compression of **both activation and activation gradient,** which differs from CompAct[1], GaLore[2], VeLORA[3], Grass[4], WeLore[5] as stated in Response 3/n.
> > >
> > > Understanding the reviewer, as well as other readers’ concerns about the comparison between our method and the optimizer gradient compression method, we provide the ablation study of our method compared to Galore[2] (compress weight gradient, optimizer state) and CompAct[1] (compress activation, weight gradient, optimizer state) in Appendix H (*line 1128* to *line 1167*) and analyze the strengths, weaknesses of these methods, and suggest the combination of INSTANT with these methods. ***We want to note that CompAct code wasn't available, so we reimplemented it based on the paper. We provided implementation in the updated supplementary.***
> > >
> > > | Partial Training | Backward MFLOPs | Activation Memory | QNLI | SST2 |
> > > | --- | --- | --- | --- | --- |
> > > | Galore-8 | 14495 | 13.50 | 82.45 | 90.47 |
> > > | Galore-32 | 14495 | 13.50 | *84.37* | *91.28* |
> > > | CompAct-8 | 7978 | 0.11 | 80.65 | 88.30 |
> > > | CompAct-32 | 8355 | 0.44 | *84.37* | *91.28* |
> > > | INSTANT-0 | 175 | 0.03 | 79.33 | 90.71 |
> > > | INSTANT-7 | 565 | 0.21 | 84.13 | 90.94 |
> > > | INSTANT-15 | 1018 | 0.42 | **84.68** | **91.63** |
> > > | Vanilla | 14495 | 13.50 | 86.12 | 91.63 |
> > >
> > > ---
> > >
> > > | Full Training | Backward MFLOPs | Activation Memory | QNLI | SST2 |
> > > | --- | --- | --- | --- | --- |
> > > | Galore-8 | 173946 | 162 | 90.44 | 91.74 |
> > > | Galore-32 | 173946 | 162 | **91.31** | 92.09 |
> > > | CompAct-8 | 95736 | 1.32 | 89.09 | 91.28 |
> > > | CompAct-32 | 100260 | 5.28 | 90.30 | 92.09 |
> > > | INSTANT-0 | 9143 | 2.83 | 89.66 | 92.22 |
> > > | INSTANT-15 | 15353 | 5.43 | 90.63 | *92.43* |
> > > | INSTANT-25 | 20753 | 8.52 | *90.79* | **93.35** |
> > > | Vanilla | 173946 | 162 | 91.43 | 93.23 |
> > >
> > > From the two tables above, we observe that GaLore consistently achieves better performance than CompAct under the same low-rank constraint. However, because GaLore primarily targets the optimizer states, its activation memory consumption and computational cost remain similar to Vanilla training. In contrast, CompAct substantially reduces activation memory by compressing activations during the forward pass. Compared to CompAct, under the same activation memory budget, INSTANT is able to save a large portion of the backward computational cost while achieving better performance.
> > >
> > > However, because CompAct also compresses the optimizer states, it explores a complementary dimension to INSTANT. In principle, the two approaches can be combined to simultaneously reduce activation memory, optimizer-state memory, and computational cost. We further believe that our calibration-based projection can, at least theoretically, be extended to compress optimizer states as well.

---

### Author Response · Authors · 2025-11-20
**Global Response to All Reviewers**

We highly appreciate the reviewers for their time and insightful comments on our work.

We also updated our manuscript following the Reviewers’ suggestions. **Modifications are marked in blue**. To make our work clear, we create this global response to summarize the reviewers’ concerns and our responses. In summary:

1. We addressed **all concerns** with respect to previous work (CompAct). Specifically, we highlight our differences in several aspects: training pipeline, target of compression, and methodology to do compression. Details in our responses to Reviewer FBNs.
2. We added extra baselines Galore, CompAct *(Appendix H, line 1129)*, ESPACE *(Appendix L.1, line 1377)*, INSTANT-random *(Appendix K.2 line 1338)*. All of them are less performant compared to our INSTANT.
3. We added additional related work (Galore, CompAct) and show our differences *(line 101 - 107 main paper)*
4. We add proof that calibration overhead is small compared to training overhead *(line 810 - 833 main paper)*
5. We provide time reports on V100 GPU *(Appendix J.2, line 1263)*
6. We provide additional results on the complex dataset (Places-365 with 1.8M images) (*Appendix L.3, line 1397 main paper*)

Again, we sincerely thank all reviewers for your time. We are keen on receiving your feedback on these modifications and continuing the discussion.

The Authors

**References:**

[1] CompAct: Compressed Activations for Memory‑Efficient LLM Training – Shamshoum et al., NAACL 2025, arXiv:2410.15352v1.

[2] GaLore: Memory‑Efficient LLM Training by Gradient Low‑Rank Projection – Zhao et al., ICML 2024, arXiv:2403.03507.

[3] VeLORA: Memory Efficient Training using Rank‑1 Sub‑space Activations – Miles et al., NeurIPS 2024 Poster, algorithm for compressing activations into 1-D subspace.

[4] Grass: Compute Efficient Low‑Memory LLM Training with Structured Sparse Gradients – Muhamed et al., arXiv:2406.17660.

[5] WeLore: Weight Low‑Rank Projection for Memory‑Efficient Fine‑Tuning – Jaiswal et al., ICLR 2025, arXiv:2407.11239.

[6] Bolei Zhou, Agata Lapedriza, Aditya Khosla, Aude Oliva, and Antonio Torralba. Places: A 10 million image database for scene recognition. IEEE transactions on pattern analysis and machine intelligence, 40(6):1452–1464, 2017.

---

### Author Response · Authors · 2025-11-28
**Kind Reminder for Reviewers' responses**

Dear Reviewers FBNs and jz4Y,

Thank you again for your thoughtful reviews! We have addressed all of your questions and concerns. We are keen on getting your feedback and address any remaining concerns within the discussion period.

We are looking forward to your response. Thank you for your time.



Kind regards,

The Authours

---

### Meta-Review · Area_Chair_qGPi · 2026-01-04

**Summary:**

This paper initially received two negative scores (with rejection and marginally below acceptance) and one positive score (with marginally above acceptance). After the rebuttal, Reviewer 9WbD indicated that he/she would increase the score to 6, as the authors have addressed most of his/her concerns. Actually, the main concerns of this paper lie in: (1) The insufficient comparison with existing methods (Reviewer FBNs) and discussion, especially on the CompAct; (2) Weak novelty and overstated contributions (Reviewer FBNs); (3) Lack of Wall-Clock Training Time on GPUs (Reviewer jz4Y, 9WbD); (4) Sensitivity of oversampling hyperparameters (Reviewer jz4Y).

The authors provided detailed experiments and explanations to address the above concerns. For the concerns from FBNs regarding the comparison with CompAct and the novelty, the authors provided the corresponding explanations in terms of the theory and method, which are well resolved according to the understanding of the Meta reviewer, despite no replies from Reviewer FBNs. In addition, the concerns from Reviewer jz4Y, 9WbD regarding the Wall-Clock Training Time on GPUs and the failure reason, are well addressed in the rebuttal.

Overall, the novelty is original and sufficient by projecting activations and gradients into low-rank subspaces and performing backpropagation in those compressed representations to accelerate the model training. The contributions are solid with comprehensive experiments. After reading the paper, reviews, rebuttal, and the author's message, the Meta reviewer recommends accepting this paper.

**Reviewer Concerns:**

The concerns about performance & GPU training time comparison, and novelty from all reviewers are well addressed.

**Reviewer Scores:**

Reviewer FBNs may increase the score if participating fully in the discussion.

---

### Decision · Program_Chairs · 2026-01-26

Accept (Poster)